# Analysis of the time course of COVID-19 cases and deaths from countries with extensive testing allows accurate early estimates of the age specific symptomatic CFR values

**Jessica E. Rothman[1]\***, **David Eidelberg[2]**, **Samantha L. Rothman[3]**, **Theodore R. Holford[4]**, **Douglas L. Rothman[5]**

1 Department of Epidemiology of Microbial Diseases, Yale University School of Public Health, New Haven, CT, United States of America, 2 Center for Neurosciences, Institute of Molecular Medicine, Northwell Health, Manhasset, New York, United States of America, 3 Departments of Mathematics and Computer Science, Tulane University, New Orleans, LA, United States of America, 4 Departments of Biostatistics, and Statistics and Data Science, Yale University School of Public Health and Yale University Graduate School of Arts and Sciences, New Haven, CT, United States of America, 5 Departments of Radiology and Biomedical Engineering, Yale University School of Medicine, New Haven, CT, United States of America

\* jessica.rothman27@gmail.com

**Data Availability Statement:** All data files are available from the Worlometer and Statista databases (https://www.worldometers.info/

## Abstract

### Background

Knowing the true infected and symptomatic case fatality ratios (IFR and CFR) for COVID-19 is of high importance for epidemiological model projections. Early in the pandemic many locations had limited testing and reporting, so that standard methods for determining IFR and CFR required large adjustments for missed cases. We present an alternate approach, based on results from the countries at the time that had a high test to positive case ratio to estimate symptomatic CFR.

### Methods

We calculated age specific (0–69, 70–79, 80+ years old) time corrected crude symptomatic CFR values from 7 countries using two independent time to fatality correction methods. Data was obtained through May 7, 2020. We applied linear regression to determine whether the mean of these coefficients had converged to the true symptomatic CFR values. We then tested these coefficients against values derived in later studies as well as a large random serological study in NYC at that time.

### Results

The age dependent symptomatic CFR values accurately predicted the percentage of the population infected as reported by two random testing studies in NYC. They also were in good agreement with later studies that estimated age specific IFR and CFR values from serological studies and more extensive data sets available later in the pandemic.

coronavirus/#countries; https://www.statista.com/
page/covid-19-coronavirus).

**Funding:** The authors received no specific funding
for this work.

**Competing interests:** The authors have declared
that no competing interests exist.

## Conclusions

We found that for regions with extensive testing it is possible to get early accurate symptomatic CFR coefficients. These values, in combination with an estimate of the age dependence of infection, allows symptomatic CFR values and percentage of the population that is infected to be determined in similar regions with limited testing.

## Introduction

Knowing the fraction of individuals infected with COVID-19 who will die or require hospitalization is critical for epidemiological modeling and public health policy for mitigating the disease. Unfortunately, it has been difficult to determine the ratio of symptomatic cases that are fatal (case fatality ratio, $CFR_{actual}$) and the fatality ratio for all infections (IFR). The CFR is the number of deaths divided by the number of symptomatic cases in a given time period, and the IFR is the number of deaths divided by the number of infected cases (i.e. cases that may or may not be symptomatic) in a given time period. The major problems in determining these ratios are accurate determination of the number of cases (symptomatic and total) and number of deaths, as well as their age dependence. Early determination during a surge in cases is made more difficult due to the need to correct for the time delay between infection and death. This delay can be up to several months leading to the reported CFR being initially several times lower than the actual CFR even if testing ascertains all symptomatic cases.

The difficulty in obtaining accurate case ascertainment early in a pandemic is demonstrated by the wide range in $CFR_{actual}$ and $IFR_{actual}$ estimates reported through early May 2020, despite sophisticated epidemiological tools being used to correct for missed cases. Based on our meta-analysis presented in the Results section, there was an over 10 fold range in $CFR_{actual}$ and $IFR_{actual}$ estimates reported from top epidemiological groups for the United States and United Kingdom [1–24]. A similar range was reported in an independent meta-analysis [1]. The combination of limited testing and the time dependence of $CFR_{crude}$ represent a major challenge for even the most sophisticated methods that try to correct for missed cases [25].

In this paper we present an alternate method, based on using data from regions with extensive testing, for determining $CFR_{actual}$ values in other regions with limited testing early in a COVID-19 outbreak. We *hypothesized* that even early in their outbreaks, countries that performed extensive testing and case tracking, had ascertained most of their symptomatic cases. We first validated, using a standard time to death correction method and a new method we introduce that does not require this correction, that accurate early calculations of the time corrected $CFR_{crude}$ ($CFR_{crudetimecorrected}$) can be obtained. We then showed by linear regression that the variation of the $CFR_{crudetimecorrected}$ values of the 7 countries we analyzed based on their very low positive to total COVID 19 test ratio could almost completely be explained by three age specific $CFR_{actual}$ values (0 to 69, 70 to 79, and 80 plus years). The values of these age specific $CFR_{actual}$ coefficients were then validated by comparison against serology studies in calculating the percent of the infected population in New York City in late April and early May, as well as comparison with $IFR_{actual}$ values calculated in several regions months after their initial COVID-19 surges.

Our findings have relevance to future outbreaks of COVID, particularly from new variants, by showing that accurate age specific $CFR_{actual}$ values can be obtained early in an outbreak even if extensive testing can only be applied in localized regions due to resource limitations. These values can then be applied to ascertain the actual number of infections and potential mortality in regions with limited testing.

## Methods

### Sources of data

Data for our final analyses were obtained from the Australian, Austrian, German, Iceland, Israeli, South Korean, and New Zealand government websites [8–12, 14, 15, 26, 27]. Data was also obtained from the New York City Department of Health website [17, 28]. We also used the data gathering sites Statista and Worldometer [26, 27] in our preliminary analyses. All analyses were done in R v4.0.1, and all plots were created using the "ggplot2" package.

### Overview of procedures

We present below an overview of the procedures performed in our analysis. Details of the procedures are then presented below.

**Procedure 1.** Using a time from infection to fatality distribution function, based on studies performed in January 2020 in China, we calculated a time corrected CFR crude value ($CFR_{crudetimecorrected}$) from the $CFR_{crude}$ time course of each country using standard methods [20, 22, 24, 29, 30]. The best $CFR_{crudetimecorrected}$ value was determined by goodness of fit to the curve.

**Procedure 2.** We then showed that similar values were obtained using a novel procedure we introduced that does not require knowing the time from infection to fatality distribution function. This method uses only the closed case $CFR_{crude}$ time course.

**Procedure 3.** The ability to accurately calculate $CFR_{crudetimecorrected}$ from very early time course data, was validated by showing that $CFR_{crudetimecorrected}$ values calculated from the full-time courses provided an excellent fit to even the very early portion of the curves.

**Procedure 4.** Using both methods to correct for the time dependence of $CFR_{crude}$ we calculated the overall and age group specific $CFR_{crudetimecorrected}$ for each of the 7 countries for the age groups 0–69 years old, 70–79 years old, and 80 years old and above.

**Procedure 5.** Using linear regression analysis, we found that the large majority of the 8.7-fold $CFR_{crudetimecorrected}$ variation between these countries could be explained by three constant $CFR_{actual}$ coefficients for the 0 to 69, 70–79 and 80–89 groups.

**Procedure 6.** We validated these coefficients by predicting the COVID-19 infected population in New York City in late April and Early May, which we found had excellent agreement with serology studies. In addition, the coefficients are shown to be in excellent agreement with values ascertained several months later after the initial COVID 19 surges had subsided in several regions.

| Definitions | |
|---|---|
| t | A given day after the start of the outbreak |
| j | Day person got infected; represents the start of a new cohort |
| C | Case: only individuals who are symptomatic |
| I | Infection: individuals who are symptomatic or asymptomatic |
| $n_C(j)$ | Number of new cases on day j |
| $N_C(t)$ | Cumulative number of cases on day t after the start of the outbreak: |
| | $N_C(t) = \sum_{j=1}^{t} n_C(j)$ |
| $n_D(j)$ | Number new fatalities (deaths) on day j |
| $N_D(t)$ | Cumulative number of fatalities (deaths) on day t after the start of the outbreak for the total number of cohorts, J: |
| | $N_D(t) = \sum_{j=1}^{J} n_{D_j}(t)$ |

| Definitions | |
|---|---|
| $N_I(t)$ | Cumulative number of infections on day t after the outbreak |
| $N_{CC}(t)$ | Cumulative number of closed cases (died or recovered) on day t |
| $N_R(t)$ | Cumulative number of recovered cases on day t |
| $CFR_{crude}$ | The uncorrected, often referred to as naïve/ crude, measured ratio of cumulative number of fatalities divided by the cumulative number of cases on a given day: |
| | $$CFR_{crude}(t) = \frac{N_D(t)}{\sum_{j=1}^{J} n_{Cj}}$$ |
| $CFR_{closedcase}$ | Same as $CFR_{crude}$ but measured using only data from closed cases (either recovered or dead) given by $[N_D(t)/N_{CC}(t)]$ |
| $CFR_{crudetimecorrected}$ | The corrected case fatality rate ($CFR_{crudetimecorrected}$) is the reported $CFR_{crude}$ corrected for the time delay between diagnosis and fatality |
| $CFR_{actual}$ | The true CFR value when all symptomatic cases are detected |
| $CFR_{crudetimecorrected}(0–69)$ | Case fatality ratio for the age group 0 to 69 years old |
| $CFR_{crudetimecorrected}(70+)$ | Case fatality ratio for the age group 70+ |
| $CFR_{crudetimecorrected}(70–79)$ | Case fatality ratio for the age group 70–79 |
| $CFR_{crudetimecorrected}(80+)$ | Case fatality ratio for the age group 80+ |
| $CFR^*_{crudetimecorrected()}$ | Contribution of an age group to the total $CFR_{crudetimecorrected}$ |
| | $CFR_{crudetimecorrected} = CFR^*_{crudetimecorrected(0–69)} + CFR^*_{crudetimecorrected(70+)}$ |
| $p(0–69)$ | Percentage of infected population between age 0 and 69. |
| $p(70–79)$ | Percentage of infected population between age 70 and 79. |
| $p(80+)$ | Percentage of infected population 80 years and older |
| $p(70+)$ | Percentage of infected population 70 years and older |
| IFR | The infection fatality ratio (IFR) given by the ratio of cumulative number of fatalities divided by the cumulative number of infected $[N_D(t)/N_I(t)]$; can only be achieved if the entire population is tested accurately. |
| $f_D(t)$ | Probability density function of fatality at t days after diagnosis |
| $F_D(t)$ | Cumulative distribution function obtained from $f_D(t)$ |

## Calculations

**Time correction of $CFR_{crude}(t)$ for the delay between diagnosis and fatality.** We used two independent methods to estimate the corrected CFR. In one method we corrected the reported $CFR_{crude}(t)$ for the time delay between diagnosis and fatality based on previously reported approaches [6, 7, 22, 24, 30–33]. In the second, we used closed case $CFR_{crude}(t)$ time courses, which does not require knowing the time to fatality distribution function.

In the first method, we implemented a time delay to fatality correction method using a time delay to death distribution function $f_D$ derived from reported log-normal fits of data obtained from China, between December and late January, of the percentage of fatalities of COVID-19 patients per day after diagnosis [22, 24, 29, 30, 33]. Data was used only from patients who were hospitalized outside of Hubei province to avoid the potential problem that adequate medical care was likely not available within the province, and especially in Wuhan, early in the outbreak [6, 32, 33]. For the cohort of cases diagnosed on day j, the $f_D$ at day t is described by,

$$f_D(t - j) = Lognormal(\log Mu, \log SD) \qquad 1$$

The calculated cumulative number of fatalities from the cohort diagnosed on day j on day t was calculated from the cumulative distribution ($F_D$) which is the integral of Eq [1] from day j

**Table 1. Reported CFR$_{crude}$, CFR$_{actual}$, and corrected IFR values for China, the United Kingdom and the United States.** The table summarizes CFR$_{crude}$ for each country region at the time of the report, calculated CFR$_{actual}$ and IFR values through early May 2020. Details are available in the cited references [2, 4, 6–14, 16, 19, 21–24, 32, 33, 40, 41]. For the USA and UK the CFR$_{crude}$ on April 15, 2020 is listed. Studies are listed by their first author or by the location of the modeling group that reported them.

| Report | CFR$_{crude}$ | (CFR$_{actual}$) | IFR | Region |
|---|---|---|---|---|
| Bendavid et al. [2] | 3.90% | 0.18%* | 0.12–0.2% | Santa Clara County, California |
| Oxford [21] | 16.7% | 0.25%* | 0.1–0.36% | United Kingdom |
| DHHS model early April 2020 | 5.0% | 0.25% | | United States |
| DHHS model mid- April 2020 | 5.0% | 0.50% | | United States |
| Ioannidis et al. [13] | 5.0% | 0.26%* | 0.13% | United States |
| CDC May 2020 [40] | 5.0% | 0.2% | | United States |
| JHU [23] | 5.0% | 0.60% | | United States |
| Pei and Shaman [18] | 5.0% | 1.1%* | 0.56% | United States |
| Modi et al. [19] | 10.20% | 1.0%* | 0.50% | New York City |
| Imperial College [3] | 16.7% | 1.8%* | 0.90% | United Kingdom |
| Mizumoto et al. [33] | 1.80% | 0.90% | | China (Hubei province) |
| Mizumoto et al. [33] | 0.43% | 0.90% | | China (outside Hubei) |
| Li et al. [41] | 3.60% | 0.90% | 0.40% | China |
| Russell et al. [24] | 3.50% | 1.10% | 0.50% | China |
| Verity et al. [22] | 3.70% | 1.38% | 0.60% | China |
| Wu et al. [32] | 4.5% | 1.40% | | China (Wuhan) |
| Wu et al. [4] | 0.85% | 0.85% | | China (outside Wuhan) |
| Hauser et al. [6] | 2.40% | 3.00% | | China (Hubei province) |
| Baud et al. [7] | 3.60% | 5.60% | | China |
| Present Work | 1.41% | 1.58% | | Australia |
| | 3.89% | 4.25% | | Austria |
| | 4.36% | 5.00% | | Germany |
| | 0.56% | 0.58% | | Iceland |
| | 1.47% | 2.16% | | Israel |
| | 2.28% | 2.65% | | South Korea |
| | 1.41% | 1.55% | | New Zealand |
| | 3.50% | 2.19%** | 1.10%** | China (Feb 11, 2020) |
| | 10.20% | 3.60%** | 1.80%** | Adults New York City (April 22, 2020) |

**: Not time corrected based on case data.

*: estimated from IFR, **: calculated from age dependent CFR coefficients from present manuscript. Abbreviations: CDC: Center for Disease Control USA; DHHS: Department of Health and Human Services, USA; Oxford: Oxford College, U.K.; Imperial College: Imperial College, U.K.

to day t multiplied by the number of new cases on day j and the corrected CFR,

$$n_{D_j}(t) = CFR_{crudetimecorrected} * n_{C_j} * F_D(t - j) \qquad 2$$

where $t > j$.

We note that Eq [2] is equivalent to a convolution integral of $f_D(t-j)$ with a delta function centered at day j with an area of CFR*$n_{Cj}$.

The value of the CFR$_{crudetimecorrected}$ was then calculated by adjusting the value of the CFR$_{crudetimecorrected}$ in Eq [2] until the calculated CFR$_{crude}$(t) on the last day of the outbreak analyzed was equal to the reported value.

**Calculation of CFR$_{crudetimecorrected}$ from closed case CFR$_{crude}$ time courses.** The second method was based on our observation that in all countries analyzed the closed case CFR (see definitions) converged to a near constant value well prior to the value of CFR$_{crude}$. A closed

case is defined as a case that has been designated as recovered or has died. The advantage of this method is that it does not require knowledge of the time to death distribution function, only that convergence has been achieved based on time course analysis. As shown in S3 Fig, provided that the median times to fatality and for recovery stay approximately constant during the outbreak, the closed case $CFR_{crude}(t)$ will converge to the final value prior to the $CFR_{crude}(t)$.

**Assessment of the sensitivity of the correction factor to the assumed input function, $f_D$.** The function $f_D$ used for the first time correction method, was based on reports of the measured onset (day of positive test) to fatality distributions for Chinese patients outside of Wuhan who were infected in December and January by Linton et al. and Mizumoto et al. [30, 33]. These investigators modeled the distributions as Log-normal functions that were corrected for right censoring (fatalities missed due to the limited patient observation time). The best fitting distributions from these sources were very similar, with Linton reporting a best fit median of 13.2 days with a 95% CI of 11.5 to 15.3 days, and Mizumoto et al. reporting a best fit median (estimated from their reported log-mean value) of approximately 13 days [30, 34, 35].

Because these results were all obtained early in the pandemic and before the final outcome of all the patients studied was known, we tested the sensitivity of our time to death correction to the range of variation in the median and shape of the published distributions. For the median (50% of fatalities have occurred) we used values of 14, 17, and 21 days to cover the full range of reports. The studies which used gamma fits reported a very similar shape of the distribution to the studies that fit the data to a lognormal distribution, equivalent to a logSD of approximately 0.50 as reported by Mizumoto [34, 35]. Goodness of fit was determined by calculating the least squares total residual by squaring the differences between our calculated $CFR_{crude}(t)$ (using the $CFR_{crudetimecorrected}$) and the reported $CFR_{crude}(t)$ values, and then summing those squares. The simulations were performed using data from Germany due to the much larger number of infected subjects, which minimizes small number statistical simulations. We found that there were relatively small variations in goodness of fit and $CFR_{crudetimecorrected}$ values calculated over the range of 14, 17, and 21 days and for each value of the median varying logSD from 0.25 to 0.75, with the best fit being for a median of 14 days and a logSD of 0.50. We then used these values in analyzing data from the other countries.

**Calculation of median and range of age dependent $CFR_{crudetimecorrected}$ values.** We calculated the values of $CFR_{crudetimecorrected}$ for the age range of 0–69, 70–79, and 80 and above ($CFR_{crudetimecorrected}(0-69)$, $CFR_{crudetimecorrected}(70-79)$, $CFR_{crudetimecorrected}(80+)$. As described below, we then validated these values using linear regression in which we plotted the age specific components of the $CFR_{crudetimecorrected}$ for each country (e.g. $CFR^*_{crudetimecorrected (70+)}$) versus the population percentage in the age range and showed that they could be fit by constant coefficients.

**Determination by linear regression of whether the range of measured age specific $CFR_{crudetimecorrected}$ values for each country could be fit by three constant age specific $CFR_{actual}$ values (0 to 69, 70 to 79, 80+).** Despite the countries examined all having a high ratio of total tests to positive cases, there was a large variation in their $CFR_{crudetimecorrected}$ values, from 0.58 to 5.0 (Table 2). To test whether this variation could be explained by constant age dependent $CFR_{actual}$ coefficients, we first performed a simple linear regression of the proportion of $CFR_{crudetimecorrected}$ due to the 70+ group range ($CFR_{crudetimecorrected}^*{}_{(70+)}$) versus the proportion of the infected population in this age for each country ($p(70+)$).

If $CFR^*_{crudetimecorrected}$ is determined by the age specific $CFR_{actual}$ coefficients, as opposed to variations in testing or other factors not related to the disease, the value of

**Table 2. Comparison of the time corrected CFR$_{crude}$ values calculated using the closed case convergence method versus the standard time to fatality time correction method [8–12, 14, 15, 26].**

| Country | Closed Case CFR | CFR$_{crudetimecorrected}$ |
|---|---|---|
| Australia | 1.58 | 1.42 |
| Austria | 4.26 | 4.20 |
| Germany | 5.02 | 5.05 |
| Iceland | 0.57 | 0.58 |
| Israel | 2.16 | 1.72 |
| New Zealand | 1.55 | 1.51 |
| South Korea | 2.65 | 2.32 |

CFR*$_{crudetimecorrected(70+)}$ is related to CFR$_{actual}$(70+) by the following relationship:

$$CFR^*_{crudetimecorrected(70+)} = CFR_{actual}(70+) * p(70+)$$ 
3

To determine how much of the variation in CFR$_{crudetimecorrected(70+)}$ between countries can be explained by a single value of CFR$_{actual}$*(70+), we calculated the R$^2$ of the least squares regression. We also compared the value of the slope to the value of CFR$_{crudetimecorrected}$(70+) determined from the mean values of the countries analyzed.

We further broke down CFR$_{crudetimecorrected(70+)}$ to understand how much of the remaining variation could be explained by using separate constant CFR$_{actual}$ coefficients for the population in the 70–79 age group and 80+ age groups respectively using Eq [4]:

$$CFR^*_{crudetimecorrected(70+)}$$
$$= CFR_{actual}(70-79) * p(70-79) + CFR_{actual}(80+) * p(80+)$$
4

To allow the goodness of fit to be shown in one graph we normalized CFR$_{crudetimecorrected(70+)}$ to the mean value of $\frac{p(80+)}{p(70+)}$ between countries of 0.40 (Table 3). The normalization used each country's measured value of CFR$_{crudetimecorrected}$(80+) and CFR$_{crudetimecorrected}$(70–79).

$$CFR^*_{crudetimecorrected(70+)A}$$

$$= CFR_{crudetimecorrected}(70-79) * \left(1 - \left(\frac{p(80+)}{p(70+)}\right)\right)$$

$$+ CFR_{crudetimecorrected}\left(80+\right) * \left(\frac{p(80+)}{p(70+)}\right)$$
5

$$CFR_{crudetimecorrected(70+)A}$$

$$= CFR_{crudetimecorrected}(70-79) * 0.60$$
6
$$+ CFR_{crudetimecorrected}(80+) * 0.40$$

**Calculation of CFR$_{actual}$ for New York and regions of China based on the age distribution of positive cases in the population and the age specific CFR$_{actual}$ values determined from the age specific CFR$_{crudetimecorrected}$ coefficients.** We calculated the CFR$_{actual}$ for New York City and regions of China (as reported by the WHO) using the following equation:

$$CFR_{actual} = CFR_{crudetimecorrected}(0-69) * p(0-69) + CFR_{crudetimecorrected}(70-79) * p(70-79)$$
$$+ CFR_{crudetimecorrected}(80+) * p(80+)$$
7

**Table 3. Age specific fractions of cases, age specific corrected CFR, and contributions of each age group to the overall corrected CFR for each country.**

| Country | p(0–69) | p(70–79) | p(70+) | p(80+) | p(80+)/ p(70+) | CFRc(80+) | CFRc(70–79) | CFRc(70+) | CFRc(0–69) |
|---|---|---|---|---|---|---|---|---|---|
| Australia | 86.11% | 10.54% | 13.89% | 3.35% | 24.09% | 27.50% | 3.95% | 9.62% | 0.28% |
| Austria | 81.51% | 8.62% | 18.49% | 9.87% | 53.37% | 24.23% | 12.62% | 18.80% | 0.95% |
| Germany | 81.12% | 8.87% | 18.88% | 10.01% | 53.00% | 31.83% | 13.06% | 23.01% | 0.81% |
| Iceland | 95.38% | 3.40% | 4.62% | 1.22% | 26.51% | 18.95% | 5.13% | 8.79% | 0.18% |
| Israel | 91.00% | 5.30% | 9.00% | 3.70% | 41.11% | 35.30% | 9.49% | 20.10% | 0.39% |
| New Zealand | 92.21% | 7.79% | 7.79% | | | | | 17.08% | 0.22% |
| South Korea | 88.90% | 6.59% | 11.10% | 4.51% | 40.59% | 28.07% | 12.05% | 18.56% | 0.63% |
| Mean | 88.03% | 7.30% | 11.97% | 5.44% | 39.78% | 27.65% | 9.38% | 16.57% | 0.49% |
| SD | 5.40% | 2.41% | 5.40% | 3.65% | 12.52% | 5.72% | 3.97% | 5.35% | 0.30% |
| 95% CI of Mean | | | | | | (21.64%, 33.65%) | (5.22%, 13.55%) | (10.95%, 22.18%) | (0.18%, 0.81%) |
| Country | CFRc*(*0–69) | CFRc*(70+) | CFRc | CFRc*(70+)/ CFRc | CFRc(70+)A | CFRc*(0–69)A | CFRc*(70+)A | CFRcA | CFRc*(70+)A/ CFRcA |
| Australia | 0.24% | 1.34% | 1.58% | 84.62% | 13.37% | 0.24% | 1.86% | 2.10% | 88.43% |
| Austria | 0.60% | 3.54% | 4.25% | 83.29% | 17.26% | 0.60% | 3.05% | 3.76% | 81.13% |
| Germany | 0.66% | 4.34% | 5.00% | 86.84% | 20.57% | 0.66% | 3.88% | 4.54% | 85.50% |
| Iceland | 0.21% | 0.37% | 0.58% | 63.82% | 10.66% | 0.21% | 0.49% | 0.70% | 70.11% |
| Israel | 0.38% | 1.81% | 2.16% | 83.75% | 19.81% | 0.38% | 1.78% | 2.16% | 82.55% |
| New Zealand | 0.22% | 1.33% | 1.55% | 85.71% | 17.08% | 0.22% | 1.33% | 1.55% | 85.71% |
| South Korea | 0.59% | 2.06% | 2.65% | 77.73% | 18.46% | 0.59% | 2.07% | 2.66% | 77.80% |
| Mean | 0.41% | 2.11% | 2.54% | 80.82% | 16.74% | 0.41% | 2.07% | 2.50% | 81.60% |
| SD | 0.20% | 1.37% | 1.57% | 8.04% | 3.55% | 0.20% | 1.11% | 1.30% | 6.14% |
| 95% CI of Mean | | | | | (14.1%, 19.4%) | | | | |

Definitions: **p()** is the proportion of the population in the relevant age group; **CFRc(80+)** is the CFR$_{crudetimecorrected}$ for cases 80 years old and above; **CFRc(70–79)** is the CFR$_{crudetimecorrected}$ for cases 70–79 years old; **CFRc(70+)** is the CFR$_{crudetimecorrected}$ for cases 70 years old and above; **CFRc(0–69)** is the CFR$_{crudetimecorrected}$ for all cases 69 years old and below; **CFRc$_{(70+)}$** is the contribution to the overall CFR$_{crudetimecorrected}$ (CFRc) from all cases 70 years old and above; **CFRc$_{(0–69)}$** is the contribution to CFR$_{crudetimecorrected}$ from all cases 69 years old and below. The **subscript A** refers to CFRc*$_{(70+)}$ values corrected to have a fraction of 40% of cases 80 years old and above. The value was chosen to match the mean from all countries except New Zealand (which has not reported this value and therefore it was assumed to be the same as the mean of the other countries). Thus CFRc*$_{(70+)A}$ is calculated by multiplying CFRc(70+)$_A$ by p(70+). Please see Eqs 5–9 for further explanation as to how values were calculated. Data was obtained from the following references [8, 9, 11, 12, 14, 15, 26, 27].

where **p()** is the proportion of the population in the relevant age groups in China or New York City. The age specific coefficients were determined from the 7 countries analyzed as described above.

**Calculation of the percentage of the adult population of New York City that has been infected with COVID-19 on April 22, 2020.** We used Eq [3] to calculate CFR$_{actual}$ for New York City using the reported percentages of cases above 0–69, 70–79, and 80+ years. Values were interpolated from the age groups reported on the New York City public health site [17, 36].

To estimate the total number of infected individuals in the population, we divided the time corrected number of fatalities by the IFR [17]. The IFR was calculated from the CFR$_{crudetimecorrected}$ values based on the assumption that the CFR$_{actual}$ was achieved in the countries analyzed. A factor of 2 was then used to convert the CFR to IFR based on reports of half of all COVID-19 cases being asymptomatic and may have escaped detection [17, 37–39].

A time correction factor ($CF_t$) of 1.74 was calculated from the new cases per day as described above. We assumed based upon a relatively constant number of tests per day over this period that the captured cases would be proportional to the total number of new cases per day in the population [17, 38, 39].

$$Number\ of\ infections = \frac{N_D(t) * CF_t}{\frac{1}{2}CFR_{actual}}$$    8

For the total number of fatalities, we used the confirmed cases to attain a minimum estimate; we then added probable fatalities for a maximum estimate. To determine the percent of the adult population infected, we then divided the maximum and minimum number of infections by the number of adults (over age 18) in New York City [38]. The adult population number was used due to the random testing not including children, who are known to have a much lower symptomatic and total infection rate than adults [8–11, 14]. We also compared our calculations with other models using their reported IFR values (Table 1) and Eq [8].

**Simulation of the closed case CFR(t).**   To understand the basis for the apparent early convergence of the closed case $CFR_{crude}$ to the $CFR_{crudetimecorrected}$ value, we calculated the cumulative number of recoveries versus day after the outbreak using the above approach for calculating cumulative fatalities (S4 Fig). Case per day data from South Korea and Germany were used in the simulations. Based on recent reports from Verity and Bi and earlier work by Ghani with SARS, the distribution function for time to recovery $f_R$ is similar to that for fatality but with a median shifted several days later and a less right skewed distribution [22, 29, 31]. Based on these reports, we used a lognormal $f_R$ with a logSD of 0.25 and examined the effect of the median shift on the convergence to the $CFR_{crudetimecorrected}$ value of closed case CFR(t) curves [22, 31].The closed case CFR(t) was calculated using the following formula,

$$closed\ case\ CFR(t) = \frac{N_D(t)}{N_R(t) + N_D(t)}$$    9

## Results

### Meta-analysis of reported IFR and CFR values for COVID 19 as of early May 2020

Table 1 presents values reported for the UK and USA from epidemiological laboratories of $CFR_{actual}$ and $IFR_{actual}$ for COVID-19 as of early May 2020. Values reported for China are also included. For the US and UK, there was a 10-fold range in reported values, and a 6-fold range for China. The Table also presents the uncorrected CFR ($CFR_{crude}$) for each country/region. For China, the UK, and USA they were up to several fold higher than the calculated values of $CFR_{actual}$ demonstrating inadequate ascertainment of total cases (Table 1) [11, 21, 22, 26].

**Increase in the reported $CFR_{crude}(t)$ versus time after the start of the outbreak in 7 countries.**   We found in all countries examined that the reported $CFR_{crude}$ increased throughout the COVID-19 outbreak. As shown in Fig 1 the value of the reported $CFR_{crude}(t)$ for Germany rose from a low value of 0.12% on March 10, 2020 to a value of 4.36% on May 7, 2020. Our estimate of the final CFR of 5.0% is shown as a dashed horizontal line. The values shown are plotted from 10 days after the first 100 cases were reported to avoid large fluctuations due to the small numbers of initial fatalities. In S2 Fig, we show that the $CFR_{crude(t)}$ versus day curves for Austria, Australia, Iceland, Israel, and New Zealand exhibited the same behavior of a large early underestimate of the final value.

**The reported closed case $CFR_{crude}$ time course converges before the $CFR_{crude}$ time course to its final value.**   We found that for the countries we examined, the closed case

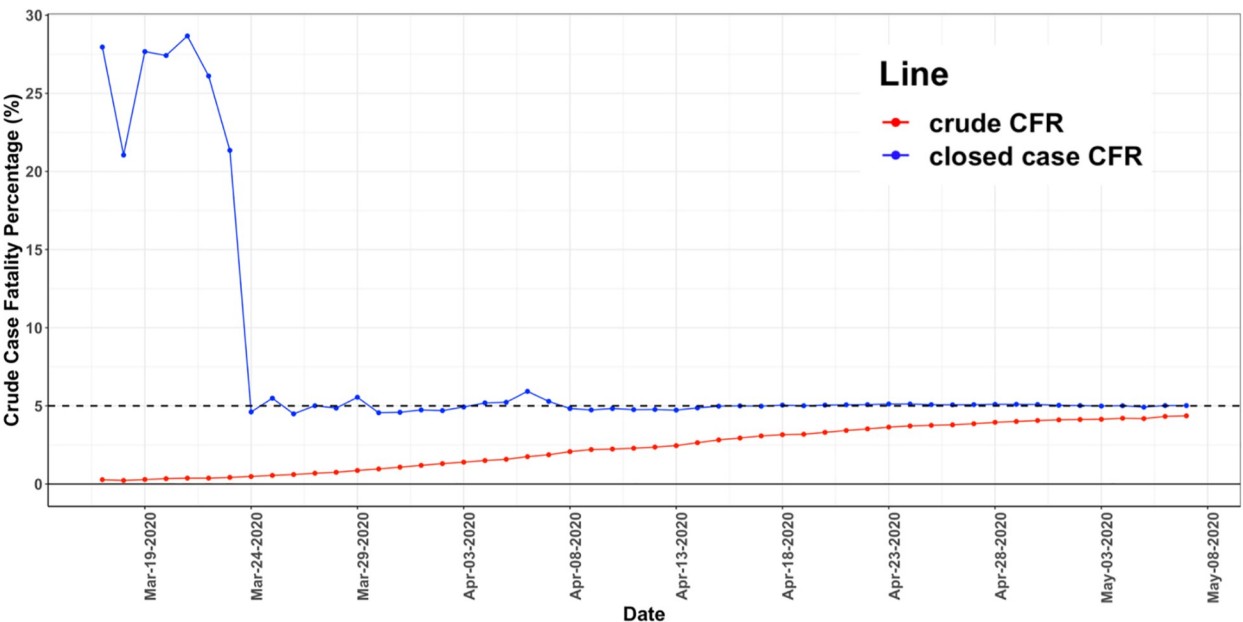

**Fig 1. $CFR_{crude}(t)$ and $CFR_{closedcase}(t)$ versus time for Germany.** The bottom curve (red) shows $CFR_{crude}(t)$ plotted versus day after outbreak. The top curve (blue) shows the same for $CFR_{closedcase}(t)$. $CFR_{crude}(t)$ increases over this period from a value of 0.12% to a value of 4.36%. It is seen that $CFR_{closedcase}(t)$ converges to the projected true of $CFR_{crude}$ earlier than the $CFR_{crude}(t)$ curve itself.

$CFR_{crude}$ value converged to a constant value prior to the $CFR_{crude}$ time course. In Fig 1, we plot $CFR_{closedcase}(t)$ and $CFR_{crude}(t)$ curves from Germany. The curves show $CFR_{crudeclosedcase}$ had converged 48 days prior to May 7, 2020, while the $CFR_{crude}$ continued to increase. S1 Fig shows that a similar convergence to a stable value also occurred for Australia, Austria, Iceland, Israel, New Zealand, and South Korea prior to convergence to its actual value at the end of the outbreak.

**Estimation of the final value of $CFR_{crude}$, using the standard time correction method and from the closed case CFR after convergence.** As shown in Table 2, the closed case CFR convergence and standard time correction methods gave similar results for all of the countries examined. This finding supports that $CFR_{closedcase}$ converged early to close to the actual $CFR_{crude}$ value.

**Assessment of the accuracy of early determination of $CFR_{crudetimecorrected}$.** To determine the accuracy of applying the time correction and closed case convergence methods early in an outbreak we simulated the $CFR_{crude}(t)$ time courses using the $CFR_{crudetimecorrecte}(t)$ values (Table 2) calculated from the entire curves. As shown in Fig 2, using the example of Germany, the curve generated using the $CFRcrude(t)$ versus time curves calculated using the $CFR_{crudetimecorrected}$ value of 5.0 (blue) matches the actual data (black) well throughout the entire time course. Similar results were found for the other countries (see SI for fits). These results demonstrate that even very early in an outbreak an accurate value of $CFR_{crude}$ can be determined.

**Determination of age specific $CFR_{actual}$ coefficients.** We calculated for each country the $CFR_{crudetimecorrected}$ coefficients in the age ranges 0 to 69, 70–79, and 80–89 (see Methods). We then tested whether the large variation in values of $CFR_{crudetimecorrected}$ between these countries could be explained by the distribution of the infected population in these age groups We chose these age ranges because of early reports that the majority of fatalities were in older age groups [24, 42]. As shown in Table 3, the age group specific values of $CFR_{crudetimecorrected}$ increased rapidly with age and were between the countries studied.

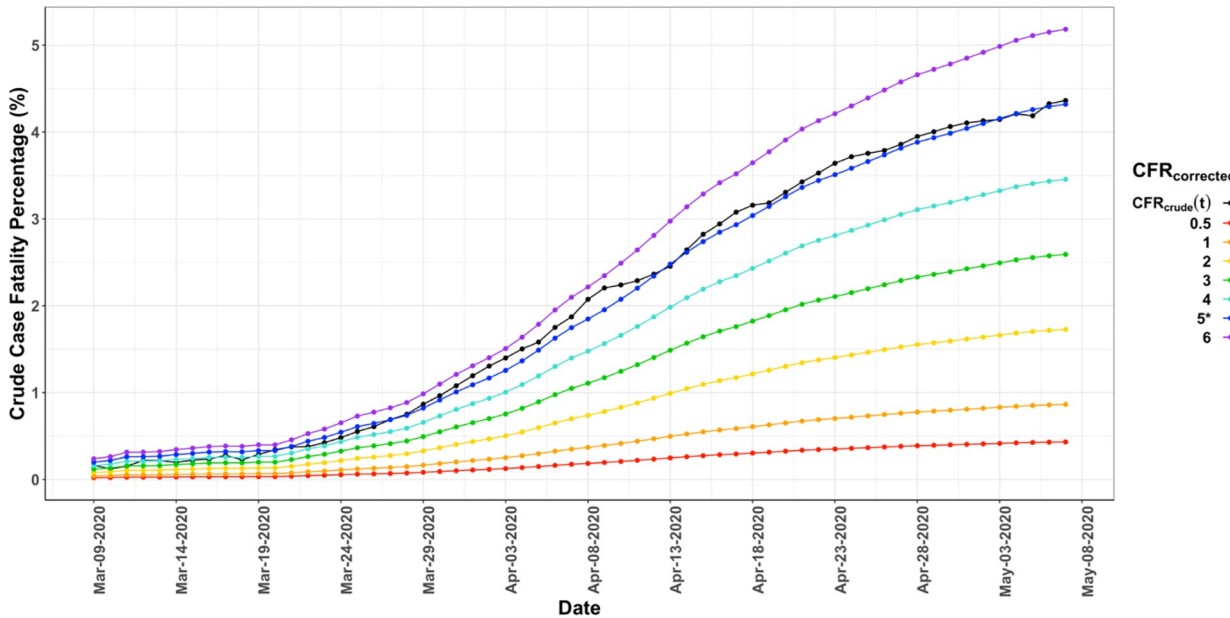

**Fig 2. Simulated and reported CFR$_{crude}$(t) versus time curves for Germany.** The reported CFR$_{crude}$(t) curve is plotted in black. Even though the reported CFR$_{crude}$(t) curve rises by more than 10-fold, it is well matched throughout the duration by the simulated CFR$_{crude}$(t) curve (blue) using the CFR$_{crudetimecorrected}$ value of 5.0% determined from the entire time course. Therefore, even early in the outbreak, when CFR$_{crude}$(t) was 10-fold lower than on May 7, 2020 (the last day used) the time correction method would have accurately predicted the true CFR$_{crude}$(t) value.

**Determination of whether case age distribution accounted for the differences in CFR$_{crudetimecorrected}$ between countries.**   Even though all of the countries studied had extensive testing there was a large variation in their overall values of CFR$_{crudetimecorrected}$ (Table 2). To determine whether this variation was due to differences in their age distribution, or other factors such as the percentage of case ascertainment, we performed a linear regression of age group specific CFR$_{crudetimecorrected}$ for each country versus the percentage of the population in the 70+ age range. In the analysis the CFR$_{crudetimecorrected}$ values calculated for each country and decomposed it into two age specific components,

$$CFR_{crudetimecorrected} = CFR^*_{crudetimecorrected(0-69)} + CFR_{crudetimecorrected(70+)} \qquad 10$$

As described in the Methods, the values of CFR$^*_{crudetimecorrected(0-69)}$ and CFR$_{crudetimecorrected(70+)}$ are related to the age specific CFR coefficients by,

$$CFR^*_{crudetimecorrected(0-69)} = CFR_{crudetimecorrected}(0-69) * p(0-69) \qquad 11$$

And

$$CFR^*_{crudetimecorrected(70+)} = CFR_{crudetimecorrected}(70+) * p(70+) \qquad 12$$

Fig 3A shows a linear regression of the term CFR$^*_{crudetimecorrected(70+)}$ plotted against the fraction of the infected population 70 years and older (blue points). The term CFR$^*_{crudetimecorrected(70+)}$ contains all deaths for cases 70 years old and above. The best fit slope corresponds to the mean value of CFR$_{crudetimecorrected}$(70+). As seen in the plot a good linearity of fit is observed with 82% of the variation explained. It is seen that for all countries the CFR$^*_{crudetimecorrected(70+)}$ term explains the large majority of CFR$_{crudetimecorrected}$ (81% +/- 8%, Table 3).

To see if the remaining variation could also be explained by age distribution we adjusted the value of CFR$_{crudetimecorrected}$(70+) measured for each country, for the fraction of their case

population 80 years and older p(80+) and taking into account the higher $CFR_{crudecorrected}$ in the 80+ group (see Methods). As seen in Fig 3B, taking into account the higher $CFR_{actual}$ of the 80+ group further improved the regression to where 89% of the variation was accounted for.

The contribution to $CFR_{crudetimecorrected}$ from cases 69 years old and younger showed a weak dependence on p(70+) (slope = 0.05, $R^2$ = 0.72), which may reflect that countries with a higher percentage of cases in the 70+ group also have a higher percentage in the 60–69 year old group which has also been shown to have an elevated risk of death from COVID-19.

**Estimation of $CFR_{actual}$ for China as of February 11, 2020 and New York City as of April 22, 2020.**   We estimated $CFR_{actual}$ for China using the mean age specific $CFR_{crudetimecorrected}$ coefficients, the case population distribution reported for China (p(0–69): 88%, p(70–79): 9%, p(80+): 3%) (39) and Eq [5] in the methods. The $CFR_{actual}$ obtained was 2.2% with a 95% CI of 1.54–2.85%. Due to the greater percentage of the infected population in the 70+ range in NYC (p(70–79): 9%, p(80+): 8%) we calculated a higher $CFR_{actual}$ value for NYC of 3.60% with a 95% CI of the mean: 2.73%-4.47%.

**Estimation from serological studies of COVID-19 from New York City of the population IFR and comparison with the calculated $CFR_{actual}$ value.**   We tested the calculated $CFR_{actual}$ for NYC against serological estimates of the percent of the adult population infected. We used the number of deaths reported in NYC as of April 22, 2020 and applied a time correction based on case per day data. We converted the $CFR_{actual}$ values to IFR values using estimates of percent asymptomatic cases from the Diamond Princess in which all passengers were tested (Methods).

The inset in Fig 4 shows our minimum and maximum calculated values (green bars) of 14.69% (95% CI of mean: 11.85%-19.43%) and 22.05% (95% CI of mean: 17.75%- 29.10%). These values are seen to be in agreement with serological studies in late April and early May that randomly tested individuals in the NYC adult population of 15.3% and 21%, respectively (blue bars) [42, 43]. In contrast the majority of reported IFR and CFR values reported up to early May 2020, predicted much higher infection percentages, as shown in the main figure.

## Discussion

Rapid determination of the actual symptomatic CFR and IFR values early in a COVID outbreak is hampered by the lag between case detection and fatality as well as incomplete case testing. To address the time lag problem, we showed that two methods provided accurate estimates of the actual $CFR_{crude}$ for COVID-19 even early in the pandemic when the reported $CFR_{crude}(t)$ was as much as 10-fold lower than the actual value. The methods were applied to 7 countries with extensive testing. We found by linear regression using the case population age distribution, that the variation in the $CFR_{crudetimecorrected}$ values could be largely explained by three constant age specific CFR coefficients. Therefore, we hypothesized that they provided an accurate estimate of age specific values of $CFR_{actual}$. The hypothesis was validated through comparison with serological testing in NYC, in which the method predicted the percent of the infected adult population more accurately than conventional methods [2, 4, 6, 13, 19, 21–24, 32–34, 41], as well as IFR calculations performed for New York City and other regions well after their initial COVID-19 surges had subsided.

To further assess the accuracy of the calculated $CFR_{actual}$ coefficients we compared them with two later studies which determined age specific IFR coefficients for NYC [25] and from a serological studies performed mainly in Europe in mid-May through early June [44] (Table 4). The Yang et al. study for NYC used a combination of advanced methods to correct for missed cases and extensive access to a wide range of data [25]. The Seoane study used data from large serological studies in multiple countries and corrected them for COVID deaths not included

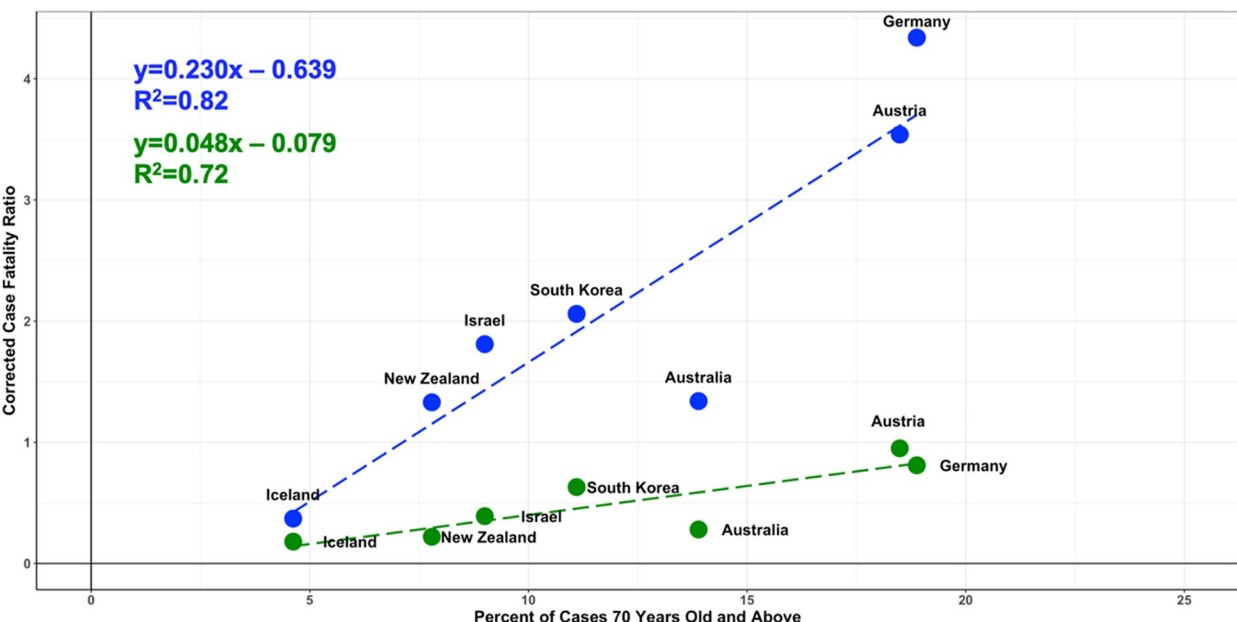

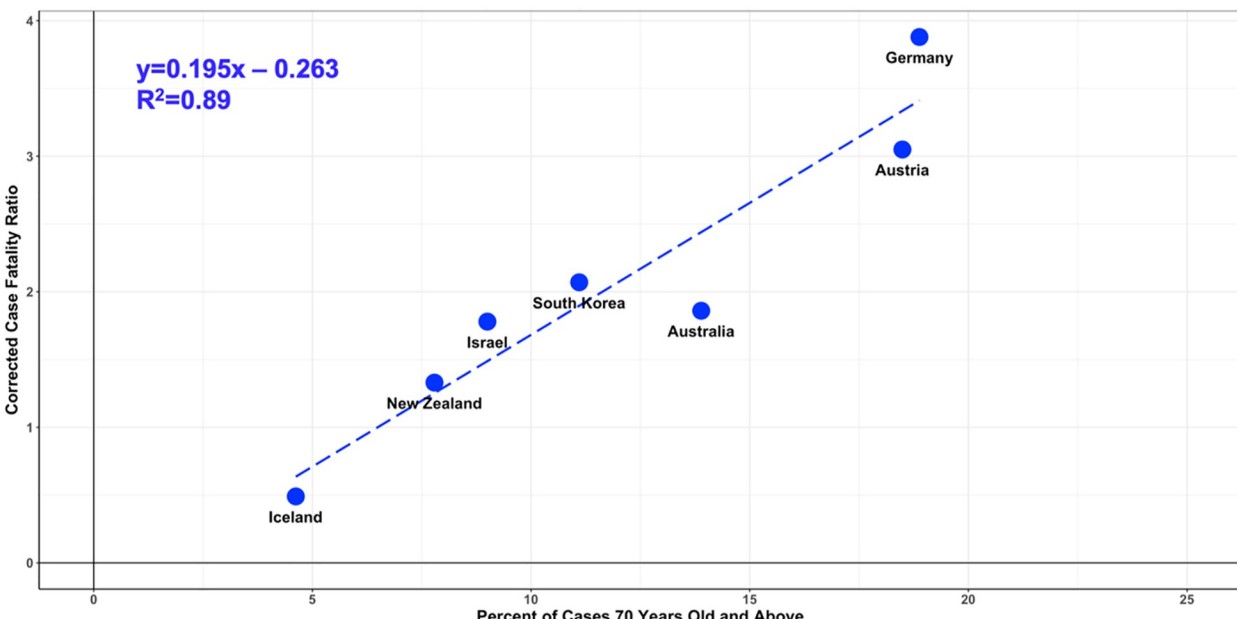

**Fig 3. Linear regression analysis of CFR\*$_{crudetimecorrected(70+)}$, CFR\*$_{crudetimecorrected(70+)A}$, and CFR\*$_{crudetimecorrected(0-69)}$ versus percent of cases 70 years old and above (p(70+)). A** shows a plot of CFR\*$_{crudetimecorrected(70+)}$ (blue) and CFR$_{crudetimecorrected(0-69)}$ (green) for each country versus the percent of cases 70 years old and above (p(70+)). It is seen that for all countries the CFR$_{crudetimecorrected(70)}$ term explains the large majority of CFR$_{crudetimecorrected}$ (81% +/- 8%). The majority of the variance in CFR$_{crudetimecorrected(70)}$ is explained by cases 70 years old and above ($R^2$ = 0.82). **B** shows a plot of CFR$_{crudetimecorrected(70+A)}$ (blue) for each country. The value of cCFR$_{70+A}$ for each country was calculated by adjusting the fraction of cases in the 70 and over group who are 80 years old and above to be 40% (p(80+)/p(70+) = 0.40), which is the mean of the countries examined (Table 3). The higher fraction of the variance explained by age for CFR$_{crudetimecorrected(70+A)}$ ($R^2$ = 0.89) indicates that the percentage of the population 80 years and older are an important factor in determining the average population value of CFR$_{crude}$.

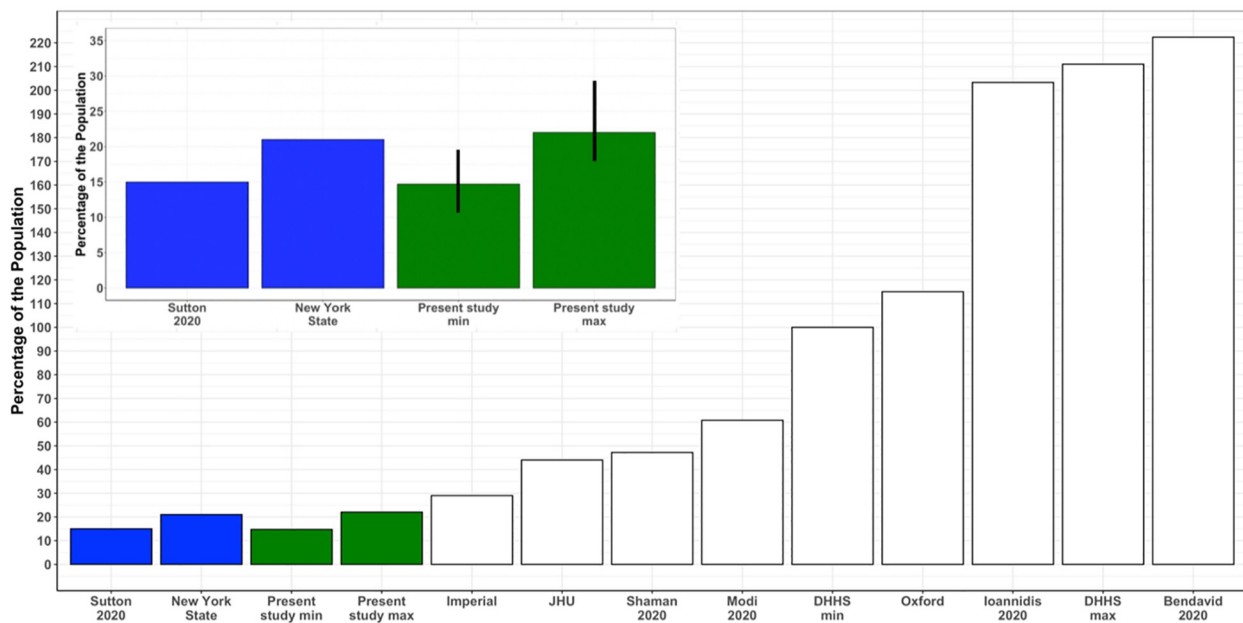

**Fig 4. Reported percentage of New York City adults infected with COVID-19 versus percentage calculated from our and other reported IFR values prior to May 7, 2020.** As shown in the inset, the predicted maximum and minimum percent of the population in New York City infected with COVID-19 is within the range determined from random adult serological testing [42, 43]. For comparison, we plotted the percentage infected using the IFR values in Table 1.

in government reported data [44]. As shown in Table 4, the age specific IFR coefficients they calculated are in excellent agreement with our findings after correction for asymptomatic cases.

There are several limitations to our study. We did not factor in preexisting conditions which has been reported as significantly affecting mortality [14, 17, 28, 38, 39, 42]. In addition, the derived age group specific $CFR_{actual}$ values may not apply to regions without advanced health care systems. However, even for medically underserved regions, our findings show that targeted high levels of testing in representative local regions could be used to rapidly determine an accurate estimate of $CFR_{actual}$. Another limitation is the need for a rapid determination of the time to fatality distribution function. However, based on our simulations, the early determinations in China were sufficient to obtain accurate $CFR_{crudetimecorrected}$ values. In addition, the closed case method does not depend upon knowing the time to fatality distribution function.

To calculate the IFR from $CFR_{actual}$, we divided the calculated $CFR_{actual}$ by a factor of 2 (50% asymptomatic) from early studies using data from the Diamond Princess [24] and Iceland [11]. This value may be an overestimate, as shown by Mizumoto, because these reports

**Table 4. Comparison of age specific IFR coefficients from the present study with serological testing studies internationally [44] and a comprehensive analysis of results from NYC [25] To facilitate comparison, we calculated a 0–64 group mean value for Yang et al. [25] and a 0–69 mean value for Seoane [44] by averaging their reported age sub group IFR values and weighting by percentage of each subgroup of the total infected population.**

| Yang et al. 2020 | | | Rothman et al. 2021 | | | Seoane 2020 | | |
|---|---|---|---|---|---|---|---|---|
| Age | Age Specific IFR | 95% CI | Age | Age Specific IFR | 95% CI | Age | Age Specific IFR | 95% CI |
| **0–64** | 0.22% | | 0–69 | 0.25% | 0.09%-0.4% | 0–69 | 0.21% | |
| **65–74** | 4.67% | 3.2%-6.7% | 70–79 | 4.70% | 2.5%-6.8% | 70–79 | 3.47% | 2.9%-4.7% |
| **75+** | 13.88% | 9.7%-17.8% | 80+ | 13.82% | 10.85%-16.8% | 80–90 | 12.70% | 11%-17% |

did not fully take into account the lag between infection and the onset of symptoms [34]. However the 50% asymptomatic estimate is still well within the present range of published values, as summarized by the latest CDC update for their best estimate values for the United States [40].

A potential confound in applying our analysis to estimate the percentage of the population infected in a region with limited testing is that the time to death correction, by both methods, assumes a constant fraction of positive case ascertainment. For NYC the validity of this assumption was supported by data from the New York City Department of Health that the number of tests per day was close to constant during the period up to April 22, 2020 and furthermore the total number of deaths reported by mid-June, at which point there would be few remaining fatalities, was similar to our projection based on time correction [17, 38, 39].

As shown in Fig 4, our calculation of the minimum and maximum percentage of the adult population in New York City that has been infected by COVID-19 agreed with the recent studies that performed random testing of segments of the adult population (Fig 4) [28, 42, 43]. In one study, 15.3% of women entering two New York City hospitals to give birth were found by testing to be infected with COVID-19 (33 out of 215 having the virus) [43]. In the second study the New York City infected population was estimated at 21%, this from 3000 serological antibody-based measurements of passersby at testing stations near public areas in New York City and other regions in New York State (with the results reported on April 22, 2020) [42]. The New York City findings were replicated from subsequent testing of 5500 cases reported April 28, 2020 (24% infected) and 15,500 cases reported on May 2, 2020 (19.9% infected). Due to the heterogeneity in COVID-19 fatalities and cases within even New York City, and due to the restricted age range of the groups examined (18–75 for the New York State study), these percent infection values may be overestimates [17, 38]. However, given that the large majority of cases in New York City are between ages 18 and 75, it is unlikely that this bias would have a large impact.

A limitation in determining IFR with serological studies is the percentage of false positives and negatives, which particularly impacts the accuracy when they are applied to region with a low percentage of infections in the population. Since the initial application of serological testing the problem false positives and negatives and how they vary between available tests has been evaluated in detail [45]. The impact of false positives was likely less significant for the New York State study because of the much high percentage of the New York City population that was infected. Additional validation of the New York results is from their finding consistently of low infection percentages (~ 1.0%) in several regions in New York State outside of the New York City metro area which supports a relatively low false positive rate in their testing [17, 28, 42]. Similarly, the serological studies from Europe were from populations with an infection percentage at least several fold higher than anticipated false positives [44].

Having early accurate age specific values CFR$_{actual}$ and IFR is vitally important for predicting the total number of cases and fatalities from COVID-19 and the impact of potential public health measures. As shown in Table 1 and Fig 4, the IFR/CFR$_{actual}$ values used in most of the leading epidemiological models in early May 2020 were not compatible with the number of infections in New York City, and this may have impacted the accuracy of projections of cases and fatalities made at that time. Our approach, in combination with targeted high testing in selected regions, has the potential to accurately determine CFR$_{actual}$ even when adequate testing is not available for the whole population.

## Supporting information

**S1 Fig. Plots of reported CFR$_{crude}$(t) and closed case CFR$_{crude}$(t) for Australia, Austria, Iceland, Israel, New Zealand, and South Korea.** Shown below are plots of the reported closed

case $CFR_{crude}(t)$ curve and reported $CFR_{crude}(t)$ curve for Austria, Australia, Iceland, Israel, New Zealand, and South Korea. The dashed gray line is the value which the closed case $CFR(t)$ has converged to. As for Germany (Fig 1), it is seen that the reported closed case $CFR(t)$ curve converges to a near constant value before the $CFR_{crude}(t)$ curve. We found (Fig 2, S2 Fig), that for all countries we examined that the converged value of the closed case CFR was close to the optimum for predicting the $CFR_{crude}(t)$ curve, consistent with it being a good approximation of the true corrected CFR for each country.
(PDF)

**S2 Fig. Plots of simulated and reported $N_D(t)$ and $CFR_{crude}(t)$ curves for Australia, Austria, Iceland, Israel, New Zealand, and South Korea.** Similar plots are presented as for Fig 2 for Germany showing the $CFR_{crude}(t)$ versus day curves for different values of the corrected CFR. In all cases a lognormal $f_D$ was used with a median value of 14 days and a logSD of 0.50. The simulated curves calculated using the closed case $CFR_{crude}$ on May 7, 2020 as the corrected CFR value are designated by an asterisk.
(PDF)

**S3 Fig. Sensitivity analysis to assess the effect of parameters of the lognormal distribution functions.** The plots below show results from the sensitivity analysis to assess the effect of the parameters of the lognormal distribution functions ($f_D$) on the simulated curves. The approximate best fit value of the corrected CFR was 5.0 (blue line asterisk) which was also the closed case CFR value on the last day plotted. The data from Germany was used for this optimization due to it having the largest number of cases of the nations studied and therefore least susceptible to statistical fluctuations. Fig 1 shows the simulated curves generated for medians of 14, 17, and 21 days and a logSD = 0.50. The effect of increasing the median resulted in the shape of the simulated curves undershooting the reported $CFR_{crude}(t)$ curve especially early in the time course due to more deaths being shifted to later dates. Decreasing the median (not shown) had the opposite effect with the simulated curves overshooting the reported data early in the time course. We also examined the effect of the logSD value on the simulated curves. Fig 2 shows the simulated curves generated for a median of 14 days and logSD values of 0.25, 0.5, and 0.75. The sensitivity logSD throughout that range was found to be low with an optimum at 0.50 which is consistent with the original reports [1, 2].
(PDF)

**S4 Fig. Simulated closed case CFR curves for Germany and South Korea.** In order to understand the basis of the early convergence of the closed case CFR we performed simulations of its time course using cases per day of from Germany and South Korea. Less information is available about the recovery distribution function than the fatality distribution function ($f_R$). Based on the study of SARS by Ghani and coworkers $f_R$ is substantially less skewed than $F_D$ [1]. This finding is consistent with the reports from early data obtained in China for COVID-19 by Bi et al. and Verity et al. who also found that the median of the $f_R$ was several days later than for $f_D$ [2, 3]. We assessed the impact of the time to recovery distribution function by simulated the closed case CFR curve using the optimum $f_R$ (median 14 days, logSD 0.50) to calculate $N_D(t)$ and $f_R$ distributions with logSD = 0.25 and median values of 14 days, 16 days, and 18 days. For input data we used the number cases per day for Germany and South Korea. The corrected CFR for each country was used in the simulations. Below we show the simulated closed case CFR curves for Germany and South Korea. Also plotted is the simulated crude CFR curve for each country. It is seen that for all of the recovery distributions evaluated the closed case CFR initially overshoots the corrected CFR value and then converges to it. The smallest overshoot and fastest convergence was for when $f_R$ had the same median value as $f_D$. In all cases the

$CFR_{crude}$ curve took longer to converge than the closed case CFR curve, consistent with the reported data from Germany and South Korea (Fig 1 and S1 Fig). The decay portion of the closed case CFR curve for South Korea was consistent with a $f_R$ median of 16 days while for Germany a 14-day median better predicted the rapid convergence to the corrected CFR values. The reported initial rise in the closed case CFR for both countries was less well predicted by the simulations, potentially due to differences in the criteria for recovery early in the outbreaks.
(PDF)

**S1 Table. Ratio of total to positive tests and tests per 1,000,000 in the population.** This table shows the ratio of negative COVID-19 tests to 1 positive COVID-19 test, and the number of COVID-19 tests per 1,000,000 in the population for each of the 7 countries included in our analysis, as of May 10, 2020 [1–8].
(PDF)

## Acknowledgments

The authors acknowledge invaluable assistance from Julia Rothman in harvesting the time course data used to perform the analysis from multiple sources. Gerard Bossard provided expert review and editing of portions of the manuscript. DLR acknowledges helpful suggestions for the paper from Gail Rothman, John Rothman, Jeff Evelhoch, Gerard Sanacora, Kevin Behar, Marcia Johnson, Barbara Gulanski and Anthony Basile.

## Author Contributions

**Conceptualization:** Jessica E. Rothman, Douglas L. Rothman.

**Data curation:** Samantha L. Rothman.

**Formal analysis:** Jessica E. Rothman, Samantha L. Rothman, Douglas L. Rothman.

**Methodology:** Jessica E. Rothman, David Eidelberg, Theodore R. Holford, Douglas L. Rothman.

**Software:** Samantha L. Rothman.

**Supervision:** Douglas L. Rothman.

**Validation:** Jessica E. Rothman, David Eidelberg, Theodore R. Holford, Douglas L. Rothman.

**Visualization:** Jessica E. Rothman, Douglas L. Rothman.

**Writing – original draft:** Jessica E. Rothman, Douglas L. Rothman.

**Writing – review & editing:** Jessica E. Rothman, David Eidelberg, Samantha L. Rothman, Theodore R. Holford, Douglas L. Rothman.

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
