## [Decision Letter · Decision Letter 0]

3 Sep 2020

PONE-D-20-16432

Analysis of the time and age dependence of the case-fatality-ratio for COVID-19 in seven countries with a high total-to-positive test ratio suggests that the true CFR may be significantly underestimated for the United States in current models

PLOS ONE

Dear Dr. Rothman,

Thank you for submitting your manuscript to PLOS ONE. After careful consideration, we feel that it has merit but does not fully meet PLOS ONE’s publication criteria as it currently stands. Therefore, we invite you to submit a revised version of the manuscript that addresses the points raised during the review process.

We look forward to receiving your revised manuscript.

Kind regards,

Lucy C. Okell

Academic Editor

PLOS ONE

Journal Requirements:

4. Please include your tables as part of your main manuscript and remove the individual files. Please note that supplementary tables (should remain/ be uploaded) as separate "supporting information" files

Reviewers' comments:

Reviewer's Responses to Questions

**Comments to the Author**

1. Is the manuscript technically sound, and do the data support the conclusions?

Reviewer #1: Yes

Reviewer #2: Partly

2. Has the statistical analysis been performed appropriately and rigorously? 

Reviewer #1: No

Reviewer #2: No

3. Have the authors made all data underlying the findings in their manuscript fully available?

Reviewer #1: No

Reviewer #2: Yes

4. Is the manuscript presented in an intelligible fashion and written in standard English?

Reviewer #1: No

Reviewer #2: Yes

5. Review Comments to the Author

Reviewer #1: I suggest readers shortening the title and let readers quickly catch the whole concept and purpose of this study at a quick glance.

The number of Tables and Figures should be condensed to report the main findings in this study. Figure 6 is nice but the computation of ture CFR should be easy to follow for non-specialist

Reviewer #2: Summary: The authors study age and time dependence of the case fatality ratio in a handful of example countries. The questions asked by the authors are certainly important. The data is presented in a useful way and many of the conclusions reached are consistent with the best available evidence for different severity estimates of COVID. However, I believe the paper has a number of methodological and presentational shortcomings. I believe the paper is not of publishable standard currently and to be frank it would need almost a complete overhaul before it would be publishable.

Major comments:

Most of the mathematics in the text I believe needs to be re-written and re-formatted for readability, as it is currently, its incredibly hard to read as it doesn't flow. It includes many uses of plain english ("CFR", age ranges, "corrected" etc), which make it almost impossible to read. On top of that, the logic that the authors are trying to move through as they describe the methods often feels clunky. Its not completely clear why certain choices are made for example.

The methods include some overstated claims I believe. For example, the subsection called "Optimization of the parameters of f_D" I do not believe is describing parameter optimisation. I could be wrong, but based on how it is currently written, it seems as though a couple of arbitrary choices were made for medians and log SDs of the distribution and inspecting the goodness of fit. So perhaps the optimal parameters out of the three or so choices of median were determined, but the section is misleading. Why were these choices made. To be frank, I believe this style of modelling is of quite poor standard

I am very surprised that there is no mention of under-ascertainment of cases as a source of bias when estimating the CFR of COVID. It is well-known that many, even most in some countries, of symptomatic cases are (or at least have been, during peaks of certain countries epidemics) under-ascertained. This can bias estimates of the CFR, along with many other important epidemiological quantities upwards significantly. The authors must mention this at the very least. If the purpose of the paper is to provide accurate CFR estimates, the authors need to adjust for this. I don't see any attempt to do this in the current form of the manuscript.

The methods are often very crude. For example, dividing by 2 to move from CFR to IFR, because roughly half of the infections were asymptomatic on the Diamond Princess. The proportion of symotomatic/asymptomatic on the Diamond Princess is known and publicly available, so using the exact ratio is very easy. Furthermore, accurate inferred estimates of this proportion and its dependency on age have been estimated, using very detailed transmission models and fitted using MCMC. I can't quite understand why accurate estimates have not been used and very crude unjustified methods have been used.

There is somewhat of a mismatch between the detail in descriptive data and some of the estimates and the crudeness of some elements of the methods. The authors are clearly aware that accuracy is required when reporting important estimates. However, they seem far less aware that the same level of accuracy and rigor is required in the methods, especially when more accurate/rigorous approaches are not much more effort or difficult (given the inordinate of high quality studies on COVID already available).

6. PLOS authors have the option to publish the peer review history of their article (what does this mean?). If published, this will include your full peer review and any attached files.

Reviewer #1: No

Reviewer #2: No

---

## [Author Response · Author response to Decision Letter 0]

11 Nov 2020

We thank the reviewers for their encouragement regarding the significance of the paper and for their many helpful comments and suggestions. We have extensively revised the text of the manuscript to focus on the use of the method for determining accurate values of the symptomatic CFR from data early in the pandemic. We have also simplified and clarified the descriptions of our methods and results throughout. The paper is also considerably shortened with several figures and text focusing on minor technical details removed. We have also updated our terminology for clarity and expanded our list of definitions. Revised sections of the manuscript are in blue print. Specific replies are given below in green. 

Comments to the Author

1. Is the manuscript technically sound, and do the data support the conclusions?

Reviewer #1: Yes

Reviewer #2: Partly

We have clarified how our conclusions follow from our results throughout. .

2. Has the statistical analysis been performed appropriately and rigorously? 

Reviewer #1: No

Reviewer #2: No

We revised our explanation of our statistical and other procedures in the Calculations and throughout. 

3. Have the authors made all data underlying the findings in their manuscript fully available?

Reviewer #1: No

Reviewer #2: Yes

Data will be made available in CSV format once the article is accepted for publication.

4. Is the manuscript presented in an intelligible fashion and written in standard English?

Reviewer #1: No 

Reviewer #2: Yes

We have clarified the procedures, data used, and conclusions based on the comments of the reviewers.

5. Review Comments to the Author

Reviewer #1: I suggest readers shortening the title and let readers quickly catch the whole concept and purpose of this study at a quick glance.

The number of Tables and Figures should be condensed to report the main findings in this study. Figure 6 is nice but the computation of ture CFR should be easy to follow for non-specialist

Following the guidance of the reviewer we have reduced the number of figures, modified the title, abstract and text, and condensed the reported main findings for clarity and to emphasize the importance of the CFR determination. 

Reviewer #2: Summary: The authors study age and time dependence of the case fatality ratio in a handful of example countries. The questions asked by the authors are certainly important. The data is presented in a useful way and many of the conclusions reached are consistent with the best available evidence for different severity estimates of COVID. However, I believe the paper has a number of methodological and presentational shortcomings. I believe the paper is not of publishable standard currently and to be frank it would need almost a complete overhaul before it would be publishable.

Thank you for your support of the significance of the wall and support for our conclusions. 

We have extensively clarified our methodological procedures and in doing so believe we have addressed the reviewers specific concerns. 

Major comments:

Most of the mathematics in the text I believe needs to be re-written and re-formatted for readability, as it is currently, its incredibly hard to read as it doesn't flow. It includes many uses of plain english ("CFR", age ranges, "corrected" etc), which make it almost impossible to read. On top of that, the logic that the authors are trying to move through as they describe the methods often feels clunky. Its not completely clear why certain choices are made for example.

We have reformatted the Calculations for readability. We have also added subscripts to all of our usage of the term CFR to make clear throughout specifically what is referred to. 

The methods include some overstated claims I believe. For example, the subsection called "Optimization of the parameters of f_D" I do not believe is describing parameter optimisation. I could be wrong, but based on how it is currently written, it seems as though a couple of arbitrary choices were made for medians and log SDs of the distribution and inspecting the goodness of fit. So perhaps the optimal parameters out of the three or so choices of median were determined, but the section is misleading. Why were these choices made. To be frank, I believe this style of modelling is of quite poor standard

We have changed the subsection title to: Assessment of the sensitivity of the correction factor to the assumed input function. As we clarify the choices of medians and SD values were taken from the studies available that measured them in patients in January and February. Because there was a range of values reported we examined the impact on the time to death CFR values we calculated (referred to now as CFRcrudetimecorrected. 

I am very surprised that there is no mention of under-ascertainment of cases as a source of bias when estimating the CFR of COVID. It is well-known that many, even most in some countries, of symptomatic cases are (or at least have been, during peaks of certain countries epidemics) under-ascertained. This can bias estimates of the CFR, along with many other important epidemiological quantities upwards significantly. The authors must mention this at the very least. If the purpose of the paper is to provide accurate CFR estimates, the authors need to adjust for this. I don't see any attempt to do this in the current form of the manuscript.

The manuscript deals with the missed case problem in two ways, which we make clearer in the text. First we deal with the problem of whether it is possible to apply a correction for the time delay to death early in the pandemic when the crude CFR can be artificially low. At the time we submitted the paper this was a major question and led to confusion regarding the whether the disease was less deadly in countries such as Germany which initially had a very low crude CFR. This same confusion exists today with the resurgence of COVID cases through the United States and Europe. 

We then show for the 6 countries the following

1. Using either a standard time to death correction or closed case CFR convergence it was possible early in the pandemic to accurately predict the true CFRcrude (CFRcrudecorrected)

2. Using a time to death distribution function derived earlier from clinical data in China was sufficiently robust to no have a large influence on the correction, as shown in the SI by our sensitivity analysis

Net we addressed the question of whether the large variation between countries in CFRcrudetimecorrected was due to differences in ascertainment or instead was due to the age dependence of CFRtrue. Based on our analysis the large majority of the variation could be explained by age which strongly supports that the individual countries were successfully ascertaining the large majority of their symptomatic cases. 

We also add a comparison of our derived age specific CFR coefficients (CFRactual in the revised manuscript) with a recent study by Shaman and coworkers published in Lancet in which they calculated from much more extensive data the IFR (and symptomatic CFR) for NYC and a paper by Seoane who calculated IFR values for many countries in Europe, as well as Asia, based on large published National random serology studies as well as ascertainment of deaths that were not originally reported. 

The methods are often very crude. For example, dividing by 2 to move from CFR to IFR, because roughly half of the infections were asymptomatic on the Diamond Princess. The proportion of symotomatic/asymptomatic on the Diamond Princess is known and publicly available, so using the exact ratio is very easy. Furthermore, accurate inferred estimates of this proportion and its dependency on age have been estimated, using very detailed transmission models and fitted using MCMC. I can't quite understand why accurate estimates have not been used and very crude unjustified methods have been used.

Since the focus of our paper was whether it was possible to calculate accurate symptomatic CFR and secondarily IFR coefficients from early data in the pandemic we used the previous correction factor as it was the best available at the time. 

We note that the correction factor was derived from several references (see text). We now report the full range reported by these references ( approximately 30 to 50% asymptomatic) and note that it is similar to the best values reported by the CDC for adults who constituted the very large majority of fatalities in the countries we obtained the coefficients from, as well as in NYC and China where we applied the coefficients to predict their IFR. 

There is somewhat of a mismatch between the detail in descriptive data and some of the estimates and the crudeness of some elements of the methods. The authors are clearly aware that accuracy is required when reporting important estimates. However, they seem far less aware that the same level of accuracy and rigor is required in the methods, especially when more accurate/rigorous approaches are not much more effort or difficult (given the inordinate of high quality studies on COVID already available).

We feel a major advantage of our method, while less sophisticated than methods that try to ascertain non ascertained cases, is it requiring just data regarding the age distribution of cases and deaths provided there is enough testing to have a high ratio of tests to positive cases. The complexity of these other methods can be a disadvantage when sufficient other data is not available, a threshold that is very hard to determine. Overall however it is clear from both our Table 1 and a more extensive meta analysis of studies at the time we performed our analysis (april through early May) that the values of the large majority of these studies often were large underestimates of what is currently the best accepted values. In contrast, as further emphasized in our new table 4, the values we determined remain very similar to best current values based on serology testing as well as more advanced case ascertainment procedures than could be applied early in the pandemic. 

 

PLOS ONE

Analysis of the time and age dependence of the case-fatality-ratio for COVID-19 in seven countries with a high total-to-positive test ratio suggests that the true CFR may be significantly underestimated for the United States in current models(Manuscript Number: PONE-D-20-16432)

Reviewer’s comments to authors:

 This is an interesting article regarding the corrected CFR calculation of COVID-19 for specific countries, particularly using the data of the delayed

time from symptom to death and the age groups. Besides the manuscript title could be shorten more for capturing the article core and major contributions to the literature, I raised several concerns below, If the title is mainly focused on the CFR calculation with delayed time and the population ages. 

 Title: Calculation of the true case fatality rate(CFR) for COVID-19 applied to infected countries using the delayed onset-to-death time and age dependence.

We have revised and shortened the title as suggested by the reviewer (with a minor modification) and emphasized this point throughout including in our revised Abstract, Introduction and Discussion. We have also eliminated redundant figures as well as text that focused on minor technical points to further sharpen the focus. 

Major concerns:

1. What are the main claims of the paper and how significant are they for the discipline?

This article attempts to propose one corrected CFR calculation to the literature. However, too many tables and Figures as well as Appendixes that make readers confused with a mess to read this article is the disadvantage. Usually, only five Tables and Figures are recommended in one article. Otherwise, the focus will be 

substantially deviated. 

We have reduced the number of figures to 4. We kept the 3 tables because we feel the information they provide is important and is best summarized in tabular form. A short fourth table was added to compare our results with age specific symptomatic CFR/IFR based on subsequent large serology studies with improved information on fatalities. 

2. Are the claims properly placed in the context of the previous literature? Have the authors treated the literature fairly?

Numerous articles have addressed and criticized the naïve CFR lower than the true CFR at the end of the epidemic. What method or model that could be applied to estimate the TRUE CFR is required to study. Even several infected countries have experienced the COVID-19 and gained the TRUE CFR, how to model the calculation of the true CFR for infected countries using the retrospective study is merit a good investigation. However, this study cannot focus the main research study and follow an appropriate approach to report the study results for this study. 

The purpose of this paper is to demonstrate that even with minimal data early on in an epidemic, our methods were still able to calculate the true symptomatic CFR as evidenced by the results of the serology studies from New York City and our comparison in Table 4 with the best values now available from large serology studies and further ascertainment of COVID deaths. 

The fact that our results are in very good agreement with these later studies, despite not having this information available supports our premise that our method, by focusing on countries that have extensive testing to determine true CFR/IFR coefficients, was effective. 

3. Do the data and analyses fully support the claims? If not, what other evidence is required?

First, one study should provide adequate data that can let the followers to mimic the study in the future. We have not seen the data regarding the age distribution in each studied countries. Furthermore, the data in Table 3, all those proportions of p(0-69). p(70-79), p(70+), and p(80+) should equal 1.0 but not in this table. For instance, the total is greater than 1.0 according to data in Australia(86.11% 10.54% 13.89% 3.35%) shown in Table 3. 

Table 3 has all of the data that the reviewer requests. We have clarified our description of the table as well as the terms used to make the information in the table easier to interpret

Second, It is not sufficient just stating data obtained from the sites (Worldometer, Statista) or data was also obtained from the New York City Department of Health website.(17,31) in Sources of data

We clarified the sources of data in Methods. Data will be made available in CSV format once the article is accepted for publication.

Third, The previous data regarding naïve CFR, age groups in population, and the time delayed in each country should be provided in this study. 

Data will be made available in CSV format once the article is accepted for publication.

4. PLOS ONE encourages authors to publish detailed protocols and algorithms as supporting information online. Do any particular methods used in the manuscript warrant such treatment? If a protocol is already provided, for example for a randomized controlled trial, are there any important deviations from it? If so, have the authors explained adequately why the deviations occurred?

The authors proposed detailed protocols and algorithms in the current study. However, if Bayes' theorem was applied in this study, the true CFR could be obtained for each infected country. As such, authors should express any difference from the Bayes’ theorem applied to compute the true CFR in Discussions. 

We have opted not to conduct this analysis in the Bayesian framework, as the methods we present were rigorous enough to estimate age specific true CFR values and predict accurate CFR/IFR values for regions with limited testing, as validated by later serology studies (table 4 and description) 

5. If the paper is considered unsuitable for publication in its present form, does the study itself show sufficient potential that the authors should be encouraged to resubmit a revised version?

The study structure should be well reorganized further. For example, following the structure of Introduction, Methods, Results, Discussions and conclusion is required. 

We had Introduction, Results, Discussion and Conclusion as sections previously, and we have added a Methods title above the “Sources of Data” header.

Moreover, the word of present is suggested using presence in the sentence of “by calculating the percent of the”. The paragraph of “In contrast, using previously reported IFR values gave minimum estimates between 1.5 and 10-fold higher (see Figure 7).(5,8–11,15,16)” is different from Table 1 with IFR values less than 1% when compared to those “between 1.5 and 10-fold” in the paragraph. 

We have extensively revised the text to make comparisons clearer 

 Why Table 1, Figure 6, Table 2 and Figure 7 are abruptly seen in Introduction instead of the number order one by one in results is concerned by readers. 

 Only Five Tables and/or Figures are suggested in this study for readers easy to capture the whole picture or concept of this study instead of so many in Figures or appendixes confusing readers. . 

We have reduced the number of figures to 4. We added one more table because it facilitated the comparison with present best IFR coefficients more effectively than having the numbers in the text 

6. Are original data deposited in appropriate repositories and accession/version numbers provided for genes, proteins, mutants, diseases, etc.?

No, we concern the data regarding the age groups in population in each country. If Table 3 provided those to readers, but the summation of proportion in age groups is greater than 1.0. As mentioned above, the the naïve CFR and age proportion for each country have been provided, the true CFR is easy to compute when Bayes’ theorem is applied.

We have clarified the age group related coefficients in the table. 

. The p() categories overlap in terms of age groups, so only certain combinations will sum to 1. We have opted not to do this analysis in the Bayesian framework. 

7. Are details of the methodology sufficient to allow the experiments to be reproduced?

No. the methodology should be easy to follow by readers. At least, I have not the feeling of easy to reproduce the study in the future.

Data will be made available in CSV format once the article is accepted for publication. 

8. Is the manuscript well organized and written clearly enough to be accessible to non-specialists?

No, the clearer and easier to read is better for readers with interest to read and cite in detail or in the future. 

We have made edits to make our wording clearer. 

Summary, I suggest readers shortening the title and let readers quickly catch the whole concept and purpose of this study at a quick glance.

 The number of Tables and Figures should be condensed to report the main findings in this study. Figure 6 is nice but the computation of ture CFR should be easy to follow for non-specialist

We thank the reviewer again for their very helpful suggestions and have incorporated the majority in the revised version

---

## [Editor Report · Decision Letter 1]

8 Jan 2021

PONE-D-20-16432R1

Analysis of the time course of COVID-19 cases and deaths from countries with extensive testing allows accurate early estimates of the age specific symptomatic CFR values

PLOS ONE

Dear Dr. Rothman,

Thank you for submitting your manuscript to PLOS ONE. After careful consideration, we feel that it has merit but does not fully meet PLOS ONE’s publication criteria as it currently stands. Therefore, we invite you to submit a revised version of the manuscript that addresses the points raised during the review process.

We look forward to receiving your revised manuscript.

Kind regards,

Wenbin Tan

Academic Editor

PLOS ONE

In summary, this article is of importance in particular comparison of CFR and IFR made among countries during the COVID-19 epidemic. For improving the readability of this article, several questions should be classified further before consideration of publication in the journal of PLOS ONE.

Major concerns:

1. It is hard to read this article for reviewers when no double spaces remained between lines.

2. Why manuscript cannot be as abstract with the formal structure of Background(or Introduction in context), Methods, Results, and Discussion(or plus condition in context). The Sources of data after Introduction is weird and rare in the format of formal scientific articles.

3. In line 3 in the Introduction, (referred to as the case fatality ratio, CFR) and infected cases (IFR) instead of infected fatality rate(IFR) used in this study.

4. At the first time to report the terms such as IFT and CFR that should be defined clearly as those in WIKI at https://en.wikipedia.org/wiki/Case_fatality_rate

5. In the Introduction, we have not been aware of the reason for conducting this study and the purpose of this study. Using Table 1 in the Introduction is rare in the scientific article. How to make a concise introduction for readers to capture the motivation and purpose of this study is of importance. For instance, using the correct CFRs with each other in countries is essential and necessary.

6. It is important for readers who are able to replicate a similar study in the future. We have not seen any that could help readers redo the study in future studies. Although authors addressed that the dataset can be provided after acceptance in the publication in response to the reviewer’s comments, the focus is not on the total fully dataset, but the calculation or process of the model parameter estimation that readers hope to understand even a partial or piece of data is enough. In this revised manuscript, I have also not been aware of any about the correct CFR calculation that can be understandable for readers.

7. In Sources of data, the definitions of symbols in equations should be followed after the formula or equation. For instance, the symbol of I for Infection: individuals who are symptomatic or asymptomatic has not found any in the formula. Keep in mind, authors should make efforts to let readers replicate and understand the study completely. Otherwise, all study are in vein. Similarly, the structure of manuscript that should be in accordance with the common style and format is necessary to be similar to (or at least not different from) other’s thinking process. For example, the section on Optimization of the parameters of fD should let us know how to obtain the fD rather than using the narrative only in the paragraph, but describing what software or statistics used in this study. That is, we are interested in this part of the study because it is important in epidemiology.

An example that can be illustrated in this study is wonderful to make readers easy capture the core of this study. It is a pity to waste the efforts made by the authors in this study because the calculation or parameter estimation is not clear and not understandable. Any result or discussion in Results and Discussion to explain the reason for the gap and difference in real-world is unnecessary.

8. I appreciate this study topic that is of interest and importance. Hopefully, this major concern can be improved in the next run of manuscript submission.

Minor concerns:

Grammar to be improved

In Abstract:

age dependent CFR time courses to explain this increase and to

- age-dependent CFT……..

Age dependent time to fatality corrected CFR was calculated using two independent

- age-dependent time to fatality……..

A linear model was developed that predicts CFR based on age dependent CFR

-age-dependent CFR……..

coefficients and the age distribution of cases. The model was tested by a linear regression of each country’s CFR against case percentage of 70 years and over.

- by linear regression

The linear model based on their age specific CFR values provides an alternative

- age-specific CFR

Introduction:

For the United States and United Kingdom, even lower true CFR and IFR values have been reported

--- the United Kingdom

reexamined the reported number of fatalities and nCFR values outbreaks on May 7, 2020 for

-2020, for

Australia, Austria, Germany, Iceland, Israel, New Zealand, and South Korea. In all cases these countries continued to have a high degree of testing and tracking and testing of contacts as

- In all cases, these……

United States and 8:1 for the United Kingdom, (Appendix 4). Therefore, their final reported

- the United Kingdom (Appendix 4)

CFR values even early in the outbreaks when the reported nCFR values were several fold lower.

-several folds lower

for the United States and United Kingdom.(5,10,13,14) Despite this range, the large

- the United Kingdom

From this model we estimated the corrected CFR for China using the linear model from the case

- From this model,

in total number of fatalities recently reported by the Chinese government.(2)

Regarding US and UK populations, there is a much wider range of CFR and IFR estimates, with

-- A total number of …….

We therefore tested our model by calculating the percent of the population who has been

-, therefore,

onset (day of positive test) to fatality distributions for Chinese patients outside of Wuhan who

A positive test

censoring (fatalities missed due to the limited patient observation time). The best fitting distributions from these sources were very similar, with Linton reporting a best fit median of

- best-fitting

13.2 days with a 95% CI of 11.5 to 15.3 days, and Mizumoto et al. reporting a best fit median

- the best fit median

all studies, based on gamma fits, was very similar, and equivalent to a logSD of approximately

--- very similar and equivalent

Goodness of fit was then determined by calculating the least squares total residual - The goodness of fit was….

the population and the age specific CFR values. Studies have reported that the nCFR - age-specific DFR

COVID-19 strongly increases with age.(5–7,9–11,15,21,22,32–34) We determined for

country the corrected CFR values for the fraction of a country’s populations of age

- for a fraction -

cumulative fatalities (Appendix 5). Case per day data from South Korea and Germany - per day of data from…..

shifted several days later and a less right skewed distribution.(1,5,32) Based on these - right-skewed distribution

population infected, we then divided the maximum and minimum number of

- and a minimum number of

due to the random testing not including children, who are known to have a much –

- , not including children,

plot using the South Korean data, which rose from a low of 0.55% on March 8, 2020 to its value

-2020,

on May 7, 2020 of 2.28%. The values shown are plotted from 10 days after the first 2020,

were reported to avoid large fluctuations, these due to the low number of fatalities - fluctuations due to the low number of

Figure 1. Reported nCFR(t) increases with time after outbreak for Germany. The nCFR the outbreak for Germany

continuously increased with time after the outbreak, from a lowest value of 0.12% on March 10, 2020 to

-2020,

its present value on May 7, 2020 of 4.36%. The dashed horizontal line at 5.0% is our -2020,

CFR from the closed case CFR value.

Figure 2. Reported nCFR(t) versus day after outbreak for South Korea. The nCFR for South

Korea is also seen to be continuously increasing from a minimum of 0.55% on March 8, 2020 to 2.28%

-2020,

curve, the closed case fatality ratio appears to be converging to a near constant value. In contrast the

a near-constant

mean of

0.129- 0.204, which is within error the same as the slope determined by linear regression value.

-the liear regression value

cCFR70+ + cCFR69. It is seen that for all countries the cCFR70 term explains the large - for all countries,

the New York City population that has been infected by COVID-19 up through April 22, 2020 in order to compare with recent studies that have performed random -0

-2020,

College model, the estimated percentages of the population infected are several fold above the

-several-fold

corrections for missed

cases.(3–11,20–22,34) We found that in all cases there was a several fold increase in -that in all cases, several-fold

n

between diagnosis and fatality. Despite the high level of testing we found a wide ---

-f testing,

When we examined the component of the CFR due to this population (cCFR70+) ---

-componence of

April 22, 2020 was 2 to 16.5-fold higher than previous values that have been applied -2020,

between 70 and 79 and 80 and over years old, but higher than the majority but not - over the years old

least several fold from the early values used to justify low estimates of the IFR for COVID-19

-several-fold

(see Figures 5A, 5B, and Appendix 3).(12,14,24–27,30) We therefore calculated the -, therefore,

We did not factor in preexisting conditions in our analysis which has been reported

- missing verb????

fraction of the case population in the 60 – 69 group which also has a highly elevated -, which also has a highly

27) However due to the low number of fatalities in several countries in the 0- 69 age -. However,

did not perform a sub analysis.

-sub-analysis.

asymptomatic. This value may be an overestimate, as shown by Mizumoto, because - by Mizumote because…..

possibility in which none of the active cases as of April 22, 2020 subsequently died ---

-2020,

COVID-19 (33 out of 215 having the virus).(16) In the second study the New York City -in the second study,

of time, and corrected for age distribution of positive cases.(11) No time correction - for the age distribution

---

## [Author Response · Author response to Decision Letter 1]

1 Mar 2021

In summary, this article is of importance in particular comparison of CFR and IFR made among countries during the COVID-19 epidemic. For improving the readability of this article, several questions should be classified further before consideration of publication in the journal of PLOS ONE.

We thank the reviewer for their careful critical reading of the paper and the support for its significance. We have made major changes to the manuscript to strengthen the paper based on the reviewer’s concerns which we summarize below.

Major concerns:

1. It is hard to read this article for reviewers when no double spaces remained between lines.

We have made the article double-spaced.

2. Why manuscript cannot be as abstract with the formal structure of Background(or Introduction in context), Methods, Results, and Discussion(or plus condition in context). The Sources of data after Introduction is weird and rare in the format of formal scientific articles.

We have moved the sources of data to the Methods section.

3. In line 3 in the Introduction, (referred to as the case fatality ratio, CFR) and infected cases (IFR) instead of infected fatality ratio (IFR) used in this study.

We make clear throughout the revised manuscript that CFR is symptomatic case fatality ratio and IFR infected fatality ratio. We also use subscripts to clarify the different uses of CFR (crude, timecorrected, actual) and provide definitions. 

4. At the first time to report the terms such as IFT and CFR that should be defined clearly as those in WIKI at https://en.wikipedia.org/wiki/Case_fatality_rate

We added the following sentence on pg 3 lines 5-8: “The CFR is the number of deaths divided by the number of symptomatic cases in a given time period, and the IFR is the number of deaths divided by the number of infected cases (i.e. cases that may or may not be symptomatic) in a given time period.” 

5. In the Introduction, we have not been aware of the reason for conducting this study and the purpose of this study. Using Table 1 in the Introduction is rare in the scientific article. How to make a concise introduction for readers to capture the motivation and purpose of this study is of importance. For instance, using the correct CFRs with each other in countries is essential and necessary.

In the revised Introduction we emphasize the reasons for conducting the study and purpose. We agree with the reviewer about how essential and necessary having accurate CFR values are. 

We establish the motivation for this study as: “. Knowing the fraction of individuals infected with COVID-19 who will die or require hospitalization is critical for epidemiological modeling and public health policy for mitigating the disease.” We further expand upon the motivation for this study on pg. 3 lines 2-20.

We included Table 1 in the introduction because we believe that the large range of reported IFR and CFR values presented demonstrate the need to develop improved approaches for rapidly determining these values without having to wait until the end of the outbreak for retrospective analysis. 

6. It is important for readers who are able to replicate a similar study in the future. We have not seen any that could help readers redo the study in future studies. 

Although authors addressed that the dataset can be provided after acceptance in the publication in response to the reviewer’s comments, the focus is not on the total fully dataset, but the calculation or process of the model parameter estimation that readers hope to understand even a partial or piece of data is enough. In this revised manuscript, I have also not been aware of any about the correct CFR calculation that can be understandable for readers.

The dataset (all data that was used in the equations) will be available as a csv. 

The revised methods section presents the complete procedures used for determining the true CFR value from reported deaths and infections per day. 

Briefly we present two methods of calculating the actual CFR from data obtained early in an outbreak: 1) a time corrected crude CFR and 2) a closed case CFR. 

 For the time corrected CFR, we used a lognormal distribution of percent deaths per day after diagnosis obtained directly from published patient studies early in the pandemic from China. The early studies were chosen because they were the only available at the time of our analysis and furthermore they provided a better test of our hypothesis that it is possible to obtain accurate corrections even early in a pandemic.

 We then evaluated the fit to the data for the countries analyzed using the range of parameters reported in the literature. We found the fits were relatively insensitive to the reported range of values. The values that gave the best fit were approximately the average of reported values with logSD of 0.5 and median days of 14 which equates LogMu(14) = 2.64 (This is given on pg. 11 lines 6-14, pg 12 lines 1-11).

 We provide the steps for this time adjustment corrected CFR (CFRcrudetimecorrected) on pg. 9 lines 12-23, pg. 10 lines 1-18. 

The second method using the closed case CFR is presented on pg. 10 lines 20-23, pg. 11 lines 1-4 and is defined as: Same as CFRcrude but measured using only data from closed cases (either recovered or dead) given by [ND(t)/NCC(t)].

7. In Sources of data, the definitions of symbols in equations should be followed after the formula or equation. For instance, the symbol of I for Infection: individuals who are symptomatic or asymptomatic has not found any in the formula. Keep in mind, authors should make efforts to let readers replicate and understand the study completely. Otherwise, all study are in vein. Similarly, the structure of manuscript that should be in accordance with the common style and format is necessary to be similar to (or at least not different from) other’s thinking process. For example, the section on Optimization of the parameters of fD should let us know how to obtain the fD rather than using the narrative only in the paragraph, but describing what software or statistics used in this study. That is, we are interested in this part of the study because it is important in epidemiology.

An example that can be illustrated in this study is wonderful to make readers easy capture the core of this study. It is a pity to waste the efforts made by the authors in this study because the calculation or parameter estimation is not clear and not understandable. Any result or discussion in Results and Discussion to explain the reason for the gap and difference in real-world is unnecessary.

We have restructured the paper to follow the standard format of Introduction, Methods, Results, and Discussion. We also extensively rewrote the paper to make the motivation for performing it clear, as well as the steps we took in the analysis and why. 

8. I appreciate this study topic that is of interest and importance. Hopefully, this major concern can be improved in the next run of manuscript submission.

We thank the reviewer for their encouragement and interest in the study and incorporated all of their very helpful suggestions. 

Minor concerns:

We have extensively rewritten the paper to address all of the concerns below. 

Grammar to be improved

In Abstract:

age dependent CFR time courses to explain this increase and to

- age-dependent CFT……..

Age dependent time to fatality corrected CFR was calculated using two independent

- age-dependent time to fatality……..

A linear model was developed that predicts CFR based on age dependent CFR

-age-dependent CFR……..

coefficients and the age distribution of cases. The model was tested by a linear regression of each country’s CFR against case percentage of 70 years and over.

- by linear regression

The linear model based on their age specific CFR values provides an alternative

- age-specific CFR

Introduction:

For the United States and United Kingdom, even lower true CFR and IFR values have been reported

--- the United Kingdom

reexamined the reported number of fatalities and nCFR values outbreaks on May 7, 2020 for

-2020, for

Australia, Austria, Germany, Iceland, Israel, New Zealand, and South Korea. In all cases these countries continued to have a high degree of testing and tracking and testing of contacts as

- In all cases, these……

United States and 8:1 for the United Kingdom, (Appendix 4). Therefore, their final reported

- the United Kingdom (Appendix 4)

CFR values even early in the outbreaks when the reported nCFR values were several fold lower.

-several folds lower

for the United States and United Kingdom.(5,10,13,14) Despite this range, the large

- the United Kingdom

From this model we estimated the corrected CFR for China using the linear model from the case

- From this model,

in total number of fatalities recently reported by the Chinese government.(2)

Regarding US and UK populations, there is a much wider range of CFR and IFR estimates, with

-- A total number of …….

We therefore tested our model by calculating the percent of the population who has been

-, therefore,

onset (day of positive test) to fatality distributions for Chinese patients outside of Wuhan who

A positive test

censoring (fatalities missed due to the limited patient observation time). The best fitting distributions from these sources were very similar, with Linton reporting a best fit median of

- best-fitting

13.2 days with a 95% CI of 11.5 to 15.3 days, and Mizumoto et al. reporting a best fit median

- the best fit median

all studies, based on gamma fits, was very similar, and equivalent to a logSD of approximately

--- very similar and equivalent

Goodness of fit was then determined by calculating the least squares total residual - The goodness of fit was….

the population and the age specific CFR values. Studies have reported that the nCFR - age-specific DFR

COVID-19 strongly increases with age.(5–7,9–11,15,21,22,32–34) We determined for

country the corrected CFR values for the fraction of a country’s populations of age

- for a fraction -

cumulative fatalities (Appendix 5). Case per day data from South Korea and Germany - per day of data from…..

shifted several days later and a less right skewed distribution.(1,5,32) Based on these - right-skewed distribution

population infected, we then divided the maximum and minimum number of

- and a minimum number of

due to the random testing not including children, who are known to have a much –

- , not including children,

plot using the South Korean data, which rose from a low of 0.55% on March 8, 2020 to its value

-2020,

on May 7, 2020 of 2.28%. The values shown are plotted from 10 days after the first 2020,

were reported to avoid large fluctuations, these due to the low number of fatalities - fluctuations due to the low number of

Figure 1. Reported nCFR(t) increases with time after outbreak for Germany. The nCFR the outbreak for Germany

continuously increased with time after the outbreak, from a lowest value of 0.12% on March 10, 2020 to

-2020,

its present value on May 7, 2020 of 4.36%. The dashed horizontal line at 5.0% is our -2020,

CFR from the closed case CFR value.

Figure 2. Reported nCFR(t) versus day after outbreak for South Korea. The nCFR for South

Korea is also seen to be continuously increasing from a minimum of 0.55% on March 8, 2020 to 2.28%

-2020,

curve, the closed case fatality ratio appears to be converging to a near constant value. In contrast the

a near-constant

mean of

0.129- 0.204, which is within error the same as the slope determined by linear regression value.

-the liear regression value

cCFR70+ + cCFR69. It is seen that for all countries the cCFR70 term explains the large - for all countries,

the New York City population that has been infected by COVID-19 up through April 22, 2020 in order to compare with recent studies that have performed random -0

-2020,

College model, the estimated percentages of the population infected are several fold above the

-several-fold

corrections for missed

cases.(3–11,20–22,34) We found that in all cases there was a several fold increase in -that in all cases, several-fold

n

between diagnosis and fatality. Despite the high level of testing we found a wide ---

-f testing,

When we examined the component of the CFR due to this population (cCFR70+) ---

-componence of

April 22, 2020 was 2 to 16.5-fold higher than previous values that have been applied -2020,

between 70 and 79 and 80 and over years old, but higher than the majority but not - over the years old

least several fold from the early values used to justify low estimates of the IFR for COVID-19

-several-fold

(see Figures 5A, 5B, and Appendix 3).(12,14,24–27,30) We therefore calculated the -, therefore,

We did not factor in preexisting conditions in our analysis which has been reported

- missing verb????

fraction of the case population in the 60 – 69 group which also has a highly elevated -, which also has a highly

27) However due to the low number of fatalities in several countries in the 0- 69 age -. However,

did not perform a sub analysis.

-sub-analysis.

asymptomatic. This value may be an overestimate, as shown by Mizumoto, because - by Mizumote because…..

possibility in which none of the active cases as of April 22, 2020 subsequently died ---

-2020,

COVID-19 (33 out of 215 having the virus).(16) In the second study the New York City -in the second study,

of time, and corrected for age distribution of positive cases.(11) No time correction - for the age distribution

---

## [Decision Letter · Decision Letter 2]

19 Apr 2021

PONE-D-20-16432R2

Analysis of the time course of COVID-19 cases and deaths from countries with extensive testing allows accurate early estimates of the age specific symptomatic CFR values

PLOS ONE

Dear Dr. Rothman,

Thank you for submitting your manuscript to PLOS ONE. After careful consideration, we feel that it has merit but does not fully meet PLOS ONE’s publication criteria as it currently stands. Therefore, we invite you to submit a revised version of the manuscript that addresses the points raised during the review process.

We look forward to receiving your revised manuscript.

Kind regards,

Wenbin Tan

Academic Editor

PLOS ONE

Reviewers' comments:

Reviewer #4: This study provides an important method to predict COVID-19 infection by calculating age specific symptomatic CFR values. However, the whole article is not well organized to show the significant finding:

1. The introduction in current status need to revised to illustrate the importance, necessity and motivation of estimates of the age specific symptomatic CFR values. Also, using table in the introduction is not a common practice for scientific papers. Please move it to result or supplementary section.

2. The procedures 1-6 is more like methods, please simplify this part and introduce the most important information or findings.

3. It’s easier for reader to detail the statistics method and software used in this study, instead of using mathematical formula. For example, what kind of regression were used to validate age dependent CFRcrudetimecorrected values.

4. The figure and table should use canonical standards for scientific articles. For example, There is no necessary to mark each day in the horizontal axis in figure 1 and 2.

---

## [Author Response · Author response to Decision Letter 2]

15 May 2021

Reviewer #4: This study provides an important method to predict COVID-19 infection by calculating age specific symptomatic CFR values. However, the whole article is not well organized to show the significant finding:

We thank the reviewer for their support of the importance of the findings. We have adopted all of their suggestions to improve the organization and overall clarity for the reader. In the revised text we use red font to indicate sections with substantial changes addressing the reviewer’s points. We also made minor changes in the Methods and Discussion for clarity.

1. The introduction in current status need to revised to illustrate the importance, necessity and motivation of estimates of the age specific symptomatic CFR values. Also, using table in the introduction is not a common practice for scientific papers. Please move it to result or supplementary section.

We have moved the description of the meta analysis and the table to the Results section.

2. The procedures 1-6 is more like methods, please simplify this part and introduce the most important information or findings.

We removed procedures 1 – 6 from the Introduction. We also modified the Introduction to end with a standard statement of the most significant findings. 

However, because a point by point overview of the procedures was requested by a previous 

reviewer we moved them to the beginning of the Methods section.

The following sections in Methods still provide detailed descriptions of each procedure. However, per the reviewer’s suggestion, we emphasize where standard methods were used. 

3. It’s easier for reader to detail the statistics method and software used in this study, instead of using mathematical formula. For example, what kind of regression were used to validate age dependent CFRcrudetimecorrected values.

We have added that all analyses were conducted in R and all plots were done in ggplot2 (pg. 7 lines 5-6) which is a standard software used in the field. 

We added “linear regression” to provide more clarity as to the type of regression (pg.12 lines 4-9). We further clarified that we used simple linear regression to validate age dependent CFRcrudetimecorrected values (Pg. 12 lines 11-22).

We however have kept the equations we used in the Methods section based upon the request of earlier reviewers, who felt that the specificity was important due to variations in the methodology within the field. 

4. The figure and table should use canonical standards for scientific articles. For example, There is no necessary to mark each day in the horizontal axis in figure 1 and 2.

We have updated Figures 1 and 2 so that every 5 days is marked instead of everyday. Also, we performed the plots in R using ggplot2 (pg. 7, lines 5 – 6) which is a standard plotting program.

---

## [Decision Letter · Decision Letter 3]

15 Jun 2021

Analysis of the time course of COVID-19 cases and deaths from countries with extensive testing allows accurate early estimates of the age specific symptomatic CFR values

PONE-D-20-16432R3

Dear Dr. Rothman,

We’re pleased to inform you that your manuscript has been judged scientifically suitable for publication and will be formally accepted for publication once it meets all outstanding technical requirements.

Kind regards,

Wenbin Tan

Academic Editor

PLOS ONE

---

## [Editor Report · Acceptance letter]

21 Jul 2021

PONE-D-20-16432R3 

Analysis of the time course of COVID-19 cases and deaths from countries with extensive testing allows accurate early estimates of the age specific symptomatic CFR values 

Dear Dr. Rothman:

I'm pleased to inform you that your manuscript has been deemed suitable for publication in PLOS ONE. Congratulations! Your manuscript is now with our production department. 

Kind regards, 

on behalf of

Dr. Wenbin Tan 

Academic Editor

PLOS ONE